# Are contributions of emissions to ozone a matter of scale? - A study using MECO(n) (MESSy v2.50)

Mariano Mertens[1], Astrid Kerkweg[2,a], Volker Grewe[1,3], Patrick Jöckel[1], and Robert Sausen[1]

[1]Deutsches Zentrum für Luft- und Raumfahrt, Institut für Physik der Atmosphäre, Oberpfaffenhofen, Germany
[2]Institut für Geowissenschaften und Meteorologie, Rheinische Friedrich-Wilhelms-Universität Bonn, Germany
[3]Delft University of Technology, Aerospace Engineering, Section Aircraft Noise and Climate Effects, Delft, the Netherlands
[a]now at: IEK-8, Forschungszentrum Jülich, Jülich, Germany

**Correspondence:** Mariano Mertens (mariano.mertens@dlr.de)

**Abstract.** Anthropogenic and natural emissions influence the tropospheric ozone budget, thereby affecting air-quality and climate. To study the influence of different emission sources on the ozone budget, often source apportionment studies with a tagged tracer approach are performed. Studies investigating air quality issues usually rely on regional models with a fine spatial resolution, while studies focusing on climate related questions often use coarsely resolved global models. It is well known that simulated ozone mixing ratios depend on the resolution of the model and the resolution of the emission inventory. Whether the contributions simulated by source apportionment approaches also depend on the model resolution, however, is still unclear. Therefore, this study firstly attempts to analyse the impact of the model, the model resolution, and the emission inventory resolution on simulated ozone contributions diagnosed with a tagging method. The differences of the ozone contributions caused by these factors are compared with differences which arise from the usage of different emission inventories. To do so we apply the MECO(n) model system which on-line couples a global chemistry-climate model with a regional chemistry-climate model equipped with a tagging scheme for source apportionment. The results of the global model (300 km horizontal resolution) are compared with the results of the regional model at 50 km (Europe) and 12 km (Germany) resolutions. Besides model specific differences and biases which are discussed in detail, our results have important implications for other modelling studies and modellers applying source apportionment methods: First, contributions of anthropogenic emissions averaged over the continental scale are quite robust with respect to the model, model resolution and emission inventory resolution. Second, the differences on the regional scale caused by different models and model resolutions can be quite large and regional models are indispensable for source apportionment studies on the sub-continental scale. Third, the difference of the contributions of stratospheric ozone transported to the surface strongly differs between the models, mainly caused by differences in the efficiency of the vertical mixing. As many models show a large difference in the downward transport of ozone to the surface, and this stratospheric ozone plays an important role for ground-level ozone it is important that source apportionment methods account for this source explicitly.

# 1 Introduction

Emissions from land transport, industry or shipping contribute largely to global budgets of trace gases like $NO_x$ and $O_3$, hereby impacting air-quality and climate (e.g. Eyring et al., 2007; Matthes et al., 2007; Hoor et al., 2009; Fiore et al., 2012; Young et al., 2013; Hendricks et al., 2017; Mertens et al., 2018). To quantify the impacts of these emissions, typically source-receptor

relationships are calculated using perturbation or source apportionment methods (e.g. Dunker et al., 2002; Emmons et al., 2012; Stock et al., 2013; Matthias et al., 2016; Huang et al., 2017; Clappier et al., 2017; Butler et al., 2018). Many studies exist quantifying the influence of anthropogenic and natural emission sources (e.g. land transport emissions, lightning) on the ozone budget, but the uncertainties of such analyses are large. Three main sources of uncertainties exist: (1) the emission inventories, (2) model biases/errors, and (3) the resolutions of the models and/or emission inventories. The influences of the first two factors,

emission inventories and model biases, have been investigated by multi-scenario and/or multi-model analyses (e.g. Eyring et al., 2007; Hoor et al., 2009; Fiore et al., 2009). Even though, the influence of the model and emission inventory resolutions on simulated ozone mixing ratios is well known (e.g. Wild and Prather, 2006; Wild, 2007; Tie et al., 2010; Holmes et al., 2014; Markakis et al., 2015), the impact of the third factor - the model and emission inventory resolutions - on the simulated contributions of specific emission sources to ozone has not yet been systematically investigated in detail. It is important to

investigate this third factor, as source apportionment studies focusing on climate usually use rather coarsely resolved global climate models (e.g. Wang et al., 1998; Lelieveld and Dentener, 2000; Grewe, 2006; Matthes et al., 2007; Dahlmann et al., 2011; Emmons et al., 2012), while air quality related studies use finer resolved regional models (e.g. Dunker et al., 2002; Li et al., 2012; Kwok et al., 2015; Valverde et al., 2016; Karamchandani et al., 2017). Therefore it is unclear, if the results from global and regional models are comparable and how large potential errors, caused by the coarse resolution of global models,

are. The present study is a first attempt to investigate the influences of the model and of the emission inventory resolutions on the ozone contributions. In detail, we investigate the influences of four different aspects on source apportionment results of ozone:

- the applied model,

- the resolution of the model,

- the resolution of the emission inventory, and

- the emission inventory.

We apply the MECO(n) (**ME**SSy-fied **ECHAM** and **CO**SMO models nested **n** times, e.g. Kerkweg and Jöckel, 2012b; Mertens et al., 2016) model system together with a detailed source apportionment method (tagging, Grewe et al., 2017). This model system couples during runtime the global chemistry-climate model EMAC (**E**CHAM5/**M**ESSy for **A**tmospheric

**C**hemistry, Jöckel et al., 2006, 2010) with the regional chemistry-climate model COSMO-CLM/MESSy (Kerkweg and Jöckel, 2012a), which consists of the COSMO-CLM model equipped with the MESSy (**M**odular **E**arth **S**ubmodel **Sy**stem, Jöckel et al., 2005, 2010) infrastructure. Due to the MESSy infrastructure, we apply identical submodels for calculating the chemical

processes as well as the same source apportionment method in the global and regional model instances. In addition, the global model instance provides consistent boundary conditions for the source apportionment to the regional model instances, allowing a detailed intercomparison of the source apportionment results on different scales. Therefore, this model system is, to our knowledge, the first available model system allowing a seamless contribution analysis from global to regional scale. With this model chain we can directly compare the results at regional and global scale, which allows us to estimate uncertainties of the contribution analyses caused by the model, the model resolution and emission inventory resolution.

This paper is organised as follows. First, Sect. 2 gives an overview of the model system, discusses the investigation strategy and the performed simulations. In Sect. 3 we present a brief evaluation of the model results against ground-level and ozone sonde observations as well as a comparison of the ozone production rates simulated by EMAC and COSMO-CLM/MESSy (Sect. 3.1). In Sect. 4 the differences in the ozone contributions caused by differences of model and emission inventory resolutions are analysed in detail. We provide a more detailed quantification of the differences for specific regions and a further discussion in Sect. 5.

## 2   Model description and experimental set-up

We apply the MECO(n) model system, which couples the global chemistry-climate model EMAC during runtime (i.e. on-line) with the regional chemistry-climate model COSMO-CLM/MESSy (Kerkweg and Jöckel, 2012b). Both models, EMAC and COSMO-CLM/MESSy, calculate the physical and chemical processes in the atmosphere and their interactions with oceans, land and human influences. They use the second version of MESSy to link multi-institutional computer codes (Jöckel et al., 2010). The core atmospheric model of EMAC is the 5th generation European Centre Hamburg general circulation model (ECHAM5, Roeckner et al., 2006). The core atmospheric model of COSMO-CLM/MESSy is the COSMO-CLM model (Rockel et al., 2008), a regional atmospheric climate model jointly further developed by the CLM-Community based on the COSMO model (Consortium for Small-scale Modelling). In the model systems acronym 'n' denotes the number of COSMO-CLM/MESSy instances nested into the global model framework. The initial and boundary conditions, which are required for each of these nested regional model instances, are provided by the next coarser resolved model instance. This model instance can either be EMAC or COSMO-CLM/MESSy. Due to the on-line coupling the boundary conditions for the regional model instances can be provided at every time step of the driving model instance. This is especially important to resolve short term variations of chemically active species. As EMAC and COSMO-CLM/MESSy calculate both, atmospheric dynamics and composition, the meteorological and chemical boundary conditions are as consistent as possible. In addition, the same chemical solver and kinetic mechanism is applied, leading to highly consistent chemical boundary conditions. Therefore, there is no need of lumping (i.e. treating different chemical species with similar chemical formula as one species), scaling of boundary conditions for specific chemical species or taking boundary conditions from different models.

More details about the MECO(n) model system are presented in a set of publications including a chemical and meteorological evaluation (Kerkweg and Jöckel, 2012a, b; Hofmann et al., 2012; Mertens et al., 2016; Kerkweg et al., 2018). The set-up of the

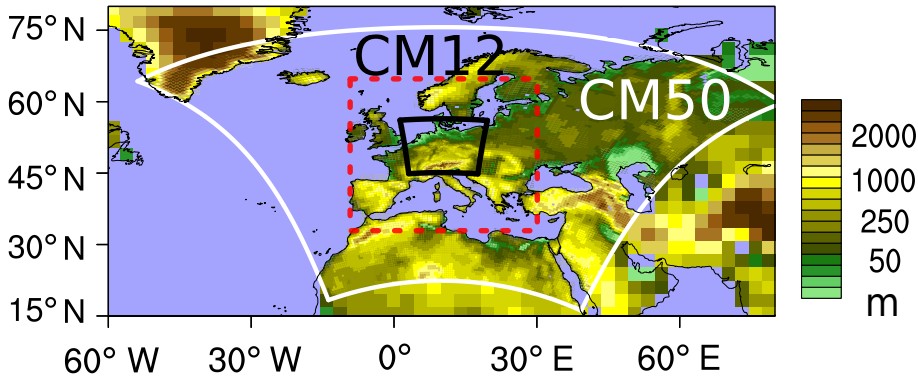

**Figure 1.** Domains of CM50 (white line) and CM12 (black line). Depicted is the topography of the continents (in m) at the resolution of the corresponding model instance. Outside the CM50 domain the topography of EMAC is displayed. Shown is the entire computational domain including the relaxation area. The dashed red square indicates the region analysed in Sect. 4. Figure is reproduced from Mertens (2017).

simulation applied in the present study is very similar to that described by Mertens et al. (2016). Therefore, we present only the most important details of the model set-up. The complete namelist set-up is part of the Supplement.

A MECO(2) set-up with one COSMO-CLM/MESSy instance over Europe with a resolution of $0.44° \times 0.44°$ ($\approx 50$ km) and one instance covering Germany with a resolution of $0.1° \times 0.1°$ ($\approx 12$ km) was applied (see Fig. 1 for the computational
domains). For simplicity, we name these two model instances hereafter CM50 and CM12. EMAC, CM50 and CM12 are running simultaneously in the same way as in externally coupled earth system models the different earth compartment model run in parallel (see Fig. 2 in Mertens et al. (2016) for details of the data exchange between the nested model instances). Both COSMO-CLM/MESSy instances use 40 vertical model levels (terrain following) with geometric height as vertical coordinate. The model top is at a height of $\approx 22$ km, the damping zone starts at 11 km height. The thickness of the lowest model layer is
$\approx 20$ m. The boundary conditions for CM50 are provided by EMAC, which is operated at a resolution of T42L31ECMWF, i.e. with a spherical truncation of T42 (corresponding to a quadratic Gaussian grid of approx. $2.8° \times 2.8°$ in latitude and longitude) with 31 hybrid pressure levels in the vertical up to 10 hPa (corresponding to $\approx 30$ km over Europe). The thickness of the lowest model layer corresponds to $\approx 60$ m over Europe. The boundary conditions for CM12 are provided by CM50. The applied MESSy version is a modified version of MESSy 2.50, including ECHAM 5.3.02 and COSMO 5.00. All changes are
included in MESSy 2.51. To facilitate a one-to-one comparison with observations, EMAC is 'nudged' by Newtonian relaxation of temperature, divergence, vorticity and the logarithm of surface pressure (Jöckel et al., 2006) towards ERA-Interim (Dee et al., 2011) reanalysis data of the years 2007 to 2010. Sea surface temperature and sea ice coverage are prescribed as boundary conditions for the simulation set-up from this data source.

Due to the MESSy infrastructure the same diagnostics or chemical process descriptions are applied in all model instances.
Following the modular structure of MESSy each diagnostic or process description is coded as a so-called submodel. The applied submodels are listed in Table 1. Besides the name of the submodel and their reference a short description provides

general information on the process or diagnostic represented by the respective submodel. Most importantly the identical kinetic solver (MECCA, Sander et al., 2011) and the identical TAGGING submodel (Grewe et al., 2017) are applied. The chemical mechanism used by MECCA considers the chemistry of ozone, methane and odd nitrogen. While alkynes and aromatics are not considered, alkenes and alkanes are considered up to $C_4$. We use the Mainz Isoprene Mechanism (MIM1, Pöschl et al., 2000) for the chemistry of isoprene and some non-methane hydrocarbons (NMHCs). The mechanisms of MECCA and for the submodel calculating the scavenging of trace gases by clouds and precipitation (SCAV, Tost et al., 2006a, 2010) are part of the supplement. The TAGGING submodel calculates the contributions of different emission sources to ozone and the relevant precursors. More details of this tagging approach are given in Sect. 2.1.

The lightning $NO_x$ emissions are calculated only in EMAC using a parametrization based on Price and Rind (1992), which is scaled to a global nitrogen oxide emission rate of $\approx 5$ Tg(N) $a^{-1}$ from flashes. In CM50 and CM12 we use the emissions from EMAC (i.e. with same geographical, vertical and temporal distribution), which are transformed on-line onto the grids of CM50 and CM12, respectively. This approach was chosen as the calculation of lightning-NOx is strongly coupled to the convection parametrisation (e.g. Tost et al., 2007). In different models and/or at different model resolutions convection occurs at different places and/or times and lightning emissions can differ largely. Our approach was chosen to allow for an easier comparison between the results of different model instances.

The calculation of emissions from soil-$NO_x$ and biogenic isoprene ($C_5H_8$) is performed by the MESSy submodel ONEMIS (described as ONLEM by Kerkweg et al., 2006b). Following the parametrizations of Yienger and Levy (1995) and Guenther et al. (1995) the respective emissions depend on the meteorological conditions. In contrast to the lightning $NO_x$ emissions, the soil-$NO_x$ and biogenic emissions are calculated separately by EMAC and CM50. This leads to differences in the soil-$NO_x$ and $C_5H_8$ emissions (see Fig. S17 in the Supplement), influencing the calculation of the contributions. We have chosen this approach, because the land sea masks differ between models and model resolutions. If the emissions calculated by EMAC are used in the COSMO-CLM/MESSy model instances, some of the emissions would occur over sea (or vice versa). This could lead to artificial errors in the contribution analyses. In EMAC, the isoprene emissions calculated by ONEMIS are scaled with a factor of 0.6 (following Jöckel et al., 2006) and in COSMO-CLM/MESSy with 0.45 (following Mertens et al., 2016).

## 2.1 Tagging for source apportionment

We apply the TAGGING submodel described by Grewe et al. (2017) for source apportionment. The tagging method is a diagnostic method, i.e. the atmospheric chemistry calculations are not influenced. Due to constraints with respect to the computational resources (e.g. computing time and memory), the detailed chemistry from MECCA is mapped on a family concept, for which the tagging is performed. The tagged species are ozone, the family of $NO_y$, the family of NMHC, CO, PAN as well as OH and $HO_2$ in a steady state approach. The tagging method itself is based on the combinatorical ansatz described by Grewe (2013). In the tagging concept the mixing ratios of the considered chemical species and families are fully decomposed into $N$ unique categories, meaning that the sum of mixing ratios over all considered categories equal the total mixing ratio of the considered species (i.e. the budget is closed):

**Table 1.** Overview of the submodels applied in EMAC and COSMO-CLM/MESSy, respectively. Both COSMO-CLM/MESSy instances use the same set of submodels. MMD* comprises the MMD2WAY submodel and the MMD library.

| Submodel | EMAC | COSMO | short description | references |
|---|---|---|---|---|
| AEROPT | x | | calculation of aerosol optical properties | Dietmüller et al. (2016) |
| AIRSEA | x | x | exchange of tracers between air and sea | Pozzer et al. (2006) |
| CH4 | x | | methane oxidation and feedback to hydrological cycle | |
| CLOUD | x | | cloud parametrisation | Roeckner et al. (2006), Jöckel et al. (2006) |
| CLOUDOPT | x | | cloud optical properties | Dietmüller et al. (2016) |
| CONVECT | x | | convection parametrisation | Tost et al. (2006b) |
| CVTRANS | x | x | convective tracer transport | Tost et al. (2010) |
| DDEP | x | x | dry deposition of aerosols and tracer | Kerkweg et al. (2006a) |
| E2COSMO | x | | additional ECHAM5 fields for COSMO coupling | Kerkweg and Jöckel (2012b) |
| GWAVE | x | | parametrisation of non-orographic gravity waves | Roeckner et al. (2003) |
| JVAL | x | x | calculation of photolysis rates | Landgraf and Crutzen (1998), Jöckel et al. (2006) |
| LNOX | x | | $NO_x$-production by lightning | Tost et al. (2007), Jöckel et al. (2010) |
| MECCA | x | x | tropospheric and stratospheric gas-phase chemistry (CCMI-base-01-tag.bat mechanism) | Sander et al. (2011), Jöckel et al. (2010) |
| MMD* | x | x | coupling of EMAC and COSMO-CLM/MESSy (including libraries and all submodels) | Kerkweg and Jöckel (2012b); Kerkweg et al. (2018) |
| MSBM | x | x | multiphase chemistry of the stratosphere | Jöckel et al. (2010) |
| OFFEMIS | x | x | prescribed emissions of trace gases and aerosols | Kerkweg et al. (2006b) |
| ONEMIS | x | x | on-line calculated emissions of trace gases and aerosols | Kerkweg et al. (2006b) |
| ORBIT | x | x | Earth orbit calculations | Dietmüller et al. (2016) |
| QBO | x | | Newtonian relaxation of the quasi-biennial oscillation (QBO) | Giorgetta and Bengtsson (1999), Jöckel et al. (2006) |
| RAD | x | | radiative transfer calculations | Dietmüller et al. (2016) |
| SCAV | x | x | wet deposition and scavenging of trace gases and aerosols | Tost et al. (2006a) |
| SEDI | x | x | sedimentation of aerosols | Kerkweg et al. (2006a) |
| SORBIT | x | x | sampling along sun synchronous satellite orbits | Jöckel et al. (2010) |
| SURFACE | x | | surface properties | Jöckel et al. (2016) |
| TAGGING | x | x | Source apportionment using a TAGGING method | Grewe et al. (2017) |
| TNUDGE | x | x | Newtonian relaxation of tracers | Kerkweg et al. (2006b) |
| TROPOP | x | x | diagnostic calculation of tropopause height and additional diagnostics | Jöckel et al. (2006) |

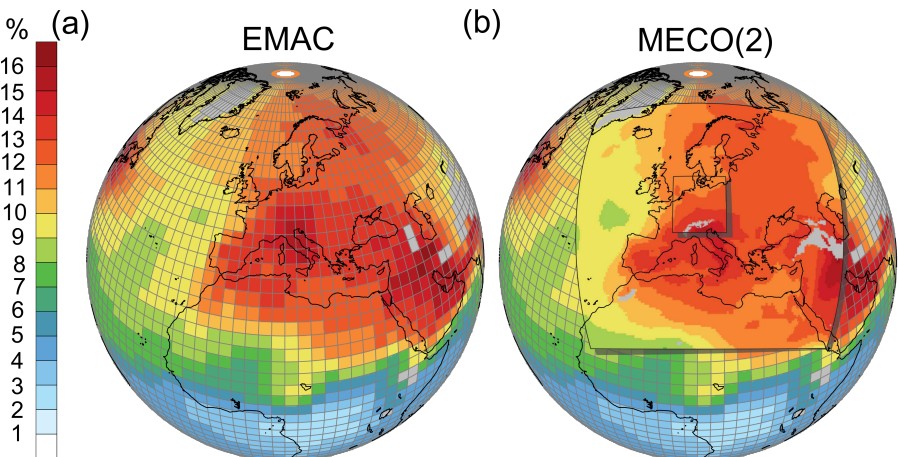

**Figure 2.** Relative contribution of land transport emissions to the ozone column up to 850 hPa (in %), averaged for July 2008; **(a)** the values calculated by the EMAC model and **(b)** the values calculated by MECO(2) with the two refinements covering Europe and Germany.

$$\sum_{\text{tag}=1}^{N} O_3^{\text{tag}} \ = \ O_3. \tag{1}$$

As an example for the generalised tagging method we consider the production of ozone by the reaction of NO with an organic peroxy radical ($RO_2$) to $NO_2$ and the organic oxy radical (RO):

$$NO + RO_2 \longrightarrow NO_2 + RO. \tag{R1}$$

According to Grewe et al. (2017) (Eq. 13 and 14 therein) this leads to the following fractional apportionment:

$$P_{R1}^{\text{tag}} \ = \tfrac{1}{2} P_{R1} \left( \frac{NO_y^{\text{tag}}}{NO_y} + \frac{NMHC^{\text{tag}}}{NMHC} \right). \tag{2}$$

$P_{R1}$ is the production rate of $O_3$ by reaction R1. $NO_y$ and NMHC are the mixing ratios of the corresponding tagged families, while species marked with $^{\text{tag}}$ represent quantities tagged for a specific category (e.g. stratosphere, land transport etc.). The denominator represents the sum of the mixing ratios over all categories of the respective tagged family/species.

Accordingly, the tagging scheme takes into account the specific reaction rates from the full chemistry scheme. Further, the fractional apportionment is inherent to the applied tagging method as due to the combinatorical ansatz every regarded chemical reaction is decomposed into all possible combinations of reacting tagged species.

The TAGGING submodel is applied in each model instance. At the lateral and top boundaries of CM50 and CM12 the tagged contributions are treated in the same manner as all chemical species, i.e. the mixing ratios of the tagged species of

the finer model instance (i.e. the absolute contributions) are relaxed towards the mixing ratios of the tagged species provided

by the driving model instance. This is depicted in Fig. 2, showing the relative contribution of the land transport emissions to ozone. EMAC calculates the contributions globally with a rather coarse resolution. With MECO(2) (Fig. 2b) the resolution over Europe and Germany is increased using the two COSMO-CLM/MESSy refinements. As the source apportionment is performed in EMAC, CM50, and CM12 - with the respective boundary conditions provided by the coarser resolved model instance - this approach allows for a consistent zooming into the area of interest within the global framework. In contrast to our approach, other tagging methods which are usually applied in regional chemistry-climate or chemistry-transport models feature no boundary conditions for the diagnosed contributions (i.e. tagged tracers) at the lateral (and top) boundaries of the regional model domain (e.g. Li et al., 2012; Kwok et al., 2015; Valverde et al., 2016). Therefore, these approaches have special categories for the contributions from lateral and/or top boundaries. In these cases long range transported ozone (or other species) are not attributed correctly to the emission sources themselves. Instead, these approaches attribute a given part of the ozone mixing ratios at a specific point to contributions from lateral and/or top boundaries. Therefore, our approach allows for a consistent zooming into the area of interest, including an apportionment of the contribution of emissions from different sources to ozone and its relevant precursors across the lateral and top boundaries of the regional model instances. Especially for chemical species with a long lifetime, such as ozone, this is important as large parts of the ozone mixing ratios at a certain place are influenced by long range transport or subsidence from the stratosphere. It is important to note that this method is a classical down-scaling method and no grid-refinement technique, which means with MECO(2) for instance over Germany we calculate the contributions three times, once in each model instance (EMAC, CM50 and CM12). By comparing the results of the different model instances the impact of the model resolution (and the model itself) can be investigated.

## 2.2 Analysis concept and performed model simulations

The goal of our study is to investigate how diagnosed contributions of different emissions to ozone in Europe are influenced by model uncertainties such as:

- the applied model,

- the resolution of the model,

- the resolution of the emission inventory, and

- the emission inventory.

For this analysis, four different MECO(2) simulations are performed which are named *REF*, *ET42*, *EBIO*, and *EVEU* (see Table 2). In all simulation the same set-up for the EMAC instance is applied, involving the MACCity emission inventory (Granier et al., 2011) with a resolution of $0.5° \times 0.5°$. The set-ups of the CM50 instance and CM12 instance (if applied) are varied systematically between the different simulations. The concept for these variations is the following.

For the *REF* simulation the MACCity is applied in EMAC, CM50 and CM12 at its finest available resolution. This means, that the MACCity emissions are transformed onto a grid of $2.8 \times 2.8°$ resolution in EMAC and to a grid of $0.44 \times 0.44°$ in CM50 (and $0.1 \times 0.1°$ resolution in CM12). The transformation from the original resolution of the emissions onto the model grid is

performed online (i.e. during runtime) via the MESSy submodel GRID (Kerkweg et al., 2018). Here, a conservative remapping approach is used to transform the emissions onto the model grid. We chose this approach, because in this way we need to store the emission data only once at their original resolution and we are in that way always using the finest possible resolution. We do not use any proxies for the downscaling of the emissions on the model grid (e.g. population density). However, due to the
different model resolutions the emissions are distributed differently into the gridboxes. The different geographical distribution of the emissions due to the transformation onto the finer grids is shown in Fig. S16 in the Supplement. This simulation serves as reference. Differences between the results of EMAC and CM50 (and CM12) can be attributed to different effects:

First, the dynamical core and physical parametrizations between EMAC and COSMO-CLM/MESSy differ, second the resolution of these models differs and third EMAC and COSMO-CLM/MESSy calculate different soil-$NO_x$ and biogenic
$C_5H_8$ emissions. The latter is due to the meteorology dependence and due to different soil types in EMAC and COSMO-CLM/MESSy.

The sensitivity simulations help to disentangle these factors. The simulation *ET42* applies the identical emissions in CM50 and in EMAC, meaning the emissions are first transformed onto the coarse grid of EMAC (2.8 x 2.8°, T42) before they are applied at this coarse resolution in CM50. Accordingly, EMAC and CM50 use the same effective resolution of the anthropogenic
emissions. By comparing the CM50 results of *REF* and *ET42*, the effect of the emission inventory resolution can be analysed.

In the simulation *EBIO* the biogenic $C_5H_8$ and soil-$NO_x$ emission as calculated by EMAC are transformed down and applied at the resolution of EMAC in CM50. By comparing the results from CM50 of *REF* and *EBIO* the effect of the differently simulated biogenic emissions can be analysed. These differences of the biogenic emissions are caused by different meteorological conditions simulated by EMAC and CM50.

Finally, the simulation *EVEU* is performed. In this simulation a different emission inventory for the emission sources shipping, land transport and anthropogenic non-traffic is used. This emission inventory is only available for Europe with a resolution of 0.0625° x 0.0625° and is an outcome of the DLR-project 'Verkehrsentwicklung und Umwelt' (VEU, Hendricks et al., 2017). The results of this simulation are used to set the differences of the contributions between the different simulations and model instances in the context of differences caused by uncertainties of the emission inventories. A full analysis of the differences
between the emission inventories is beyond the scope of the present manuscript and presented by Mertens et al. (2019). Further, the finer resolution of the emission inventory allows to compare the results of CM50 and CM12 to investigate the effect of increased model and emission inventory resolution. The total emissions applied in all simulations are given in the Supplement in Table S3 to Table S11.

The simulated period of the *REF* simulation covers 07/2007 to 12/2010. All sensitivity simulations are branched off in
12/2007 from the *REF* simulation. The simulation period of the *EVEU* simulation ranges from 12/2007 to 12/2010. The simulations *ET42* and *EBIO* cover just on year ending in 12/2008. Due to the high computational resources needed for the CM12 model instance, the CM12 instance is only activated for the period May to August 2008 and only for the simulations *REF* and *EVEU* (see also Fig. S15).

All chemical species, as well as the tagging diagnostics, are initialised from a 6-month spin-up simulation with EMAC
only (period 01/2007–07/2007). This spin-up simulation was initialised with trace gas mixing ratios from the *RC1SD-base-*

**Table 2.** Overview of the applied MECO(2) simulation set-ups and simulation periods. For the EMAC instance the same set-up is applied in all simulations, but the set-ups of the COSMO-CLM/MESSy instances (CM50 and CM12) are varied systematically. More details are given in the text. The note 'calculated by EMAC' in the row 'biogenic emissions' means that the emissions, which are calculated by EMAC, are transformed to the COSMO-CLM grid during runtime via the MMD2WAY submodel.

| Simulation | | EMAC | | CM50/CM12 | |
|---|---|---|---|---|---|
| acronym | period | anthropogenic emissions | biogenic emissions | anthropogenic emissions | biogenic emissions |
| *REF* | 07/2007-12/2010 | | | MACCity, $0.5° \times 0.5°$ | on-line calculated |
| *ET42* | 12/2007-12/2008 | MACCity, $2.8° \times 2.8°$ | on-line calculated | MACCity, $2.8° \times 2.8°$ | on-line calculated |
| *EBIO* | 12/2007-12/2008 | | | MACCity, $0.5° \times 0.5°$ | calculated by EMAC |
| *EVEU* | 12/2007-12/2010 | | | VEU, $0.0625° \times 0.0625°$ | on-line calculated |

*10a* simulation described in detail by Jöckel et al. (2016). The soil-model TERRA of COSMO-CLM/MESSy is initialised with an output of a simulation without chemistry for the period 01/1983–07/2007. MECO(n) is operated in the so called quasi chemistry transport model mode (QCTM-mode, Deckert et al., 2011; Mertens et al., 2016). In this mode chemistry and dynamics are decoupled to increase the signal-to-noise ratio for small chemical perturbations. This means, that even though the emissions differ between the different simulations, each model instance (EMAC, CM50 and CM12) simulates the same meteorology in all simulations, which does of course not imply that the meteorology between the different model instances (EMAC, CM50 and CM12) is the same. In EMAC the QCTM mode is implemented by applying the following climatologies: (a) for all radiatively active substances ($CO_2$, $CH_4$, $O_3$, $N_2O$, CFC-11 and CFC-12) for the radiation calculations, (b) nitric acid for the heterogeneous chemistry calculations (submodel MSBM (Multiphase Stratospheric Box Model) and (c) for OH, $O^1D$ and Cl for methane oxidation in the stratosphere (submodel CH4). In COSMO-CLM/MESSy only the climatology of nitric acid for the calculation of heterogeneous chemistry is needed. The applied climatologies are monthly mean values from the *RC1SD-base-10a* simulation.

For our comparison we focus on the period June–August (JJA) where the ozone production is largest. Further, we compare the results on the coarsest grid, to analyse if the finer resolution leads to any added value compared to the coarse resolution.

## 3 Model evaluation

To evaluate the performance of the different model instances and of the different simulations, we compare the model results with ground-level observations of ozone and measurements from ozone sondes. For the evaluation we use observations by the European Monitoring and Evaluation Programme (EMEP, http://www.emep.int, Tørseth et al., 2012) and ozone sonde data from the world ozone database (WOUDC, http://woudc.org). The methodology is described in detail by Mertens et al. (2016). In comparison to Mertens et al. (2016), however, we here focus on average values for June to August 2008 instead of June and December 2008. A list of the used observation data is part of the Supplement (Sect. S4).

**Table 3.** Root-mean-square error (RMSE, in µg m$^{-3}$ and normalized mean-bias error (MBE, in %) of O$_3$ for EMAC and CM50 in comparison to ground-level observations. Shown are the averaged values for June to August 2008. The values are calculated from monthly mean values. The model values are height corrected as discussed in detail by Mertens et al. (2016).

| | RMSE (in µg m$^{-3}$) | | MB (in %) | |
|---|---|---|---|---|
| | EMAC | CM50 | EMAC | CM50 |
| *REF* | | 25.2 | | 19.5 |
| *EVEU* | | 22.7 | | 16.4 |
| *ET42* | 19.6 | 26.0 | 13.1 | 20.5 |
| *EBIO* | | 26.1 | | 20.4 |

For a quantitative evaluation we chose the metrics RMSE (root mean square error) and MB (normalised mean bias error). The definition of both quantities is given in Appendix 1. Table 3 lists the RMSE and MB of the EMAC and CM50 instances for all simulations. As the EMAC set-up is identical in all simulations the model results do not change. Generally, the models results are in agreement with the measurements. The RMSE is in the range of around 19 to 26 µg m$^{-3}$ and the MB in the range of 13 to 21 %. These deviations from the measurements are in the range of comparable model systems (e.g. Knote et al., 2011; Stock et al., 2014). As already noted by Mertens et al. (2016), CM50 exhibits a larger positive ozone bias than EMAC. This bias is mainly caused by a more efficient vertical mixing in COSMO-CLM, as well as by a less stable boundary layer during night. The latter is a common problem of many models leading to diurnal cycles with too large ozone values during night, resulting in an overall ozone bias (e.g. Travis and Jacob, 2019). The results of CM12 are not presented here, as the domain covers only Germany and therefore less stations can used for evaluation. The RMSE and MB for CM50 and CM12 are given in the Supplement (Table S2), taking into account the measurements at all stations located in the region covered by the CM12 domain.

In general, CM50 simulates larger ozone mixing ratios than EMAC over the continent (see Fig. 3). This ozone bias of CM50 compared to EMAC is neither caused by the finer resolution of the emissions, nor by the different biogenic emissions compared to EMAC, because also for the results of *ET42* and *EBIO* CM50 shows a positive ozone bias compared to EMAC. Only over the Mediterranean sea, CM50 simulates less ozone compared to EMAC. These lower ozone mixing ratios can partly be attributed to the coarser resolution of the emissions in EMAC compared to CM50, as the difference is lower in the *ET42* simulation (Fig. 3b). The simulated ozone mixing ratios of CM50 are up to 7.5 nmol mol$^{-1}$ larger (JJA 2008) in *ET42* compared to *REF*. Averaged over the area of the Mediterranean sea the increase of ozone is around 3 nmol mol$^{-1}$. The application of the soil-NO$_x$ and biogenic emissions calculated by EMAC in CM50 (*EBIO*) leads to an increase of the ozone mixing ratios by 1 to 3 nmol mol$^{-1}$. The differences are largest in South Eastern Europe, the Mediterranean Sea and over the Iberian Peninsula (Fig. 3c). Overall, the differences of the results of CM50 between *REF*, *EBIO* and *ET42* are small compared to the bias between EMAC and CM50. Especially the positive ozone bias over Serbia and Bulgaria cannot be attributed to different biogenic emissions or the coarser resolution of the emission inventories in EMAC compared to CM50.

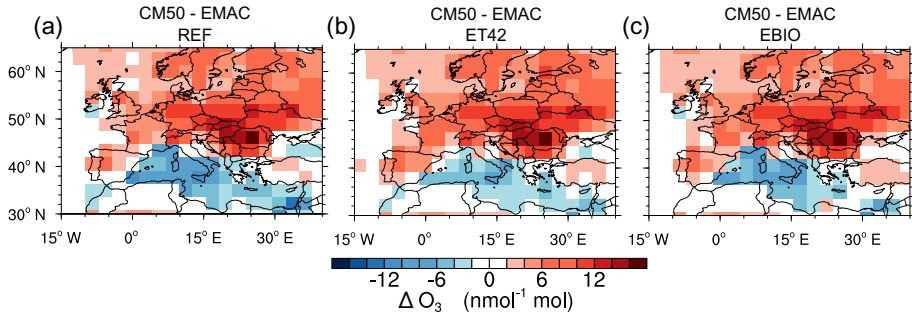

**Figure 3.** Difference between JJA 2008 averaged ozone mixing ratios (in $\mathrm{nmol\ mol^{-1}}$) as simulated by CM50 and EMAC ('CM50 MINUS EMAC'); (a) *REF* simulation, (b) *ET42* and (c) *EBIO* simulation.

Figure 4 shows scatter plots comparing observed and simulated ozone monthly mean concentrations at all considered stations of the EMEP network. The simulated concentrations for all model instances and simulations lie, with one outlier, around a factor of two of the measurements. As already discussed, the simulated ozone concentrations at most stations show a positive ozone bias. Only at some stations the simulated ozone concentrations are lower as the measured ozone concentrations. The

ozone bias is very similar in all CM50 simulations, *EBIO* and *ET42* show almost the same bias as *REF*. Only the simulation *EVEU* shows a slightly lower positive ozone bias. Accordingly, the change of the anthropogenic emission inventory has a larger impact on the model results as the influence of the emission inventory resolution and the geographical distribution of the biogenic emissions.

To evaluate the simulated ozone mixing ratios in the free troposphere, the model results are compared to ozone sonde data

(see Sect. S4 in the Supplement for a list of considered stations). In total, 510 individual ozone sonde launches are considered for the year 2008. To compare the ozone sonde data with the model results, the vertical ozone profiles simulated by the model were sampled on-line at every time-step of the model at the location were the ozone sonde was launched. Drifts of the ozone sonde by winds are not taken into account. For every launched ozone sonde, we averaged the simulated vertical profiles in time over the measurement period (usually some hours). These vertical profiles of simulated ozone mixing ratios are compared

to the measurements of the ozone sonde data. As the main focus of this comparison is the free troposphere, we restrict this analysis to all data in the pressure range of 600 to 200 $\mathrm{hPa}$.

The probability density functions (PDFs) for the measured and simulated vertical ozone distributions are displayed in Fig. 5. The results show that in the free troposphere both model instances (EMAC, CM50) simulate a very similar vertical ozone distributions. Accordingly, the positive ozone bias of CM50 compared to EMAC is confined to the boundary layer. Further, in

general a positive ozone bias is apparent, which is will known for EMAC (e.g. Righi et al., 2015; Jöckel et al., 2016).

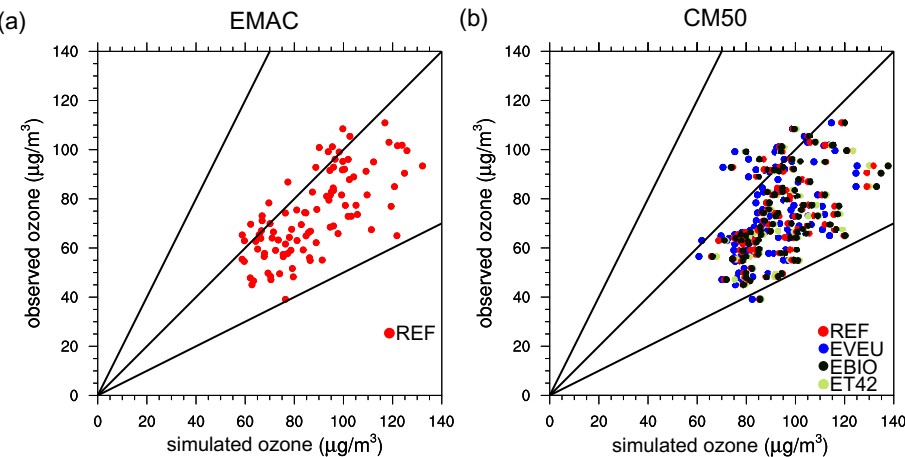

**Figure 4.** Scatter plot of the observed versus simulated ozone concentrations (in μg m$^{-3}$) for (a) EMAC and (b) CM50. Each dot represents a monthly mean value for one station in the period June to August 2008. The black lines indicate the 1:1 (observed and simulated concentrations are equal) line, and the range of a factor of two. For EMAC only the results of the *REF* simulation are shown, as the set-up of EMAC is identical in all simulations.

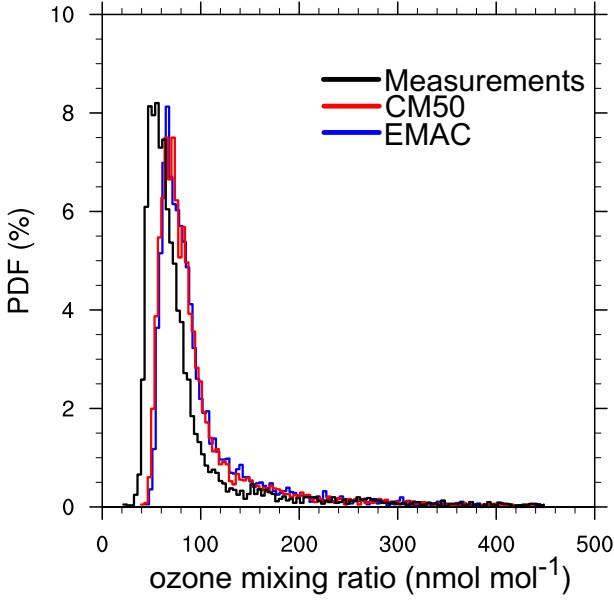

**Figure 5.** Probability density functions (PDFs) of observed (ozone sondes) and simulated vertical ozone mixing ratios in the height region between 600 and 200 hPa. Considered are 510 ozone sonde launches for 2008 in Europe.

### 3.1 Differences in ozone production

In a next step, the difference of the ozone production simulated by EMAC and CM50 is analysed (for the *REF* simulation). For this, we consider the net ozone production ($P_{O3}$) defined as:

$$P_{O3} = ProdO3 - LossO3, \tag{3}$$

5      with the production ($ProdO3$) and loss rates ($LossO3$) as diagnosed by the chemical solver (for more details see Supplement of Grewe et al. (2017)).

We define $\Delta P_{O3}$ as $\Delta P_{O3} = P_{O3}{}^{CM50} - P_{O3}{}^{EMAC}$. $\Delta P_{O3}$ is largest in the lower troposphere (see Fig. 6a). As indicated by the negative numbers, CM50 simulates in general lower values of $P_{O3}$ than EMAC. Zonally averaged $P_{O3}$ is around 60 to 80 $\mathrm{fmol\,mol^{-1}\,s^{-1}}$ lower in CM50 as in EMAC, which corresponds to 10 to 20 %. The largest differences (up to 10   100 $\mathrm{fmol\,mol^{-1}\,s^{-1}}$ or 40 %) are simulated over the Mediterranean Sea (see also Fig. S1 in the Supplement).

To separate effects caused by the emission inventory resolution from the effects caused by the model resolution and specific model biases, Fig. 6b shows the differences of $\Delta P_{O3}$ between *ET42* and *REF* ($\Delta P_{O3}{}^{ET42} - \Delta P_{O3}{}^{REF}$). The positive values indicate the effect of increased $P_{O3}$ with reduced resolution of the emission inventory, which is caused by the dilution effect of the emissions on the coarse grid (e.g. Tie et al., 2010). The differences are largest in the Mediterranean area with an increase 15   of $P_{O3}$ in CM50 of up to 40 $\mathrm{fmol\,mol^{-1}\,s^{-1}}$ in *ET42* compared to *REF*. These differences are mainly simulated in the areas of the Alboran Sea and Balearic Sea, as well as in the areas of the Levantine Sea (see also Fig. S2 in the Supplement). The main reason for these differences are the dilution of the shipping emissions, and the large anthropogenic emissions in Israel if coarse emissions are applied. As the ozone production is strongly non-linear this dilution of the emissions leads to an artificial increase of the ozone production rate.

20   The differences, which cannot be attributed directly to the resolution of the anthropogenic emission inventory, are caused by a variety of other model factors which cannot be disentangled in detail. The most important factor in this context is the enhanced vertical mixing in CM50 compared to EMAC, mainly in the boundary layer, but also due to stronger convective up- and downdraft massfluxes in CM50 compared to EMAC. The enhanced vertical mixing transports higher amounts of ozone from the free troposphere into the boundary layer, leading to higher ozone mixing ratios in the boundary layer. In addition, 25   ozone precursors are transported more efficiently from the boundary layer into the free troposphere. Further, differences in the land use classes between EMAC and CM50 lead to differences of the calculated dry deposition velocities, which affects also ozone mixing ratios near the surface (see also Mertens et al., 2016).

## 4 Contributors to ozone in Europe

Figure 7 shows the absolute and relative contributions of different emission sources to the European ozone column up to 30   850 hPa as simulated by EMAC and CM50 for the *REF* simulation (see Table S1 in the Supplement for detailed definition of the tagging categories). The largest absolute and relative ozone contributors are the anthropogenic non-traffic and the biogenic

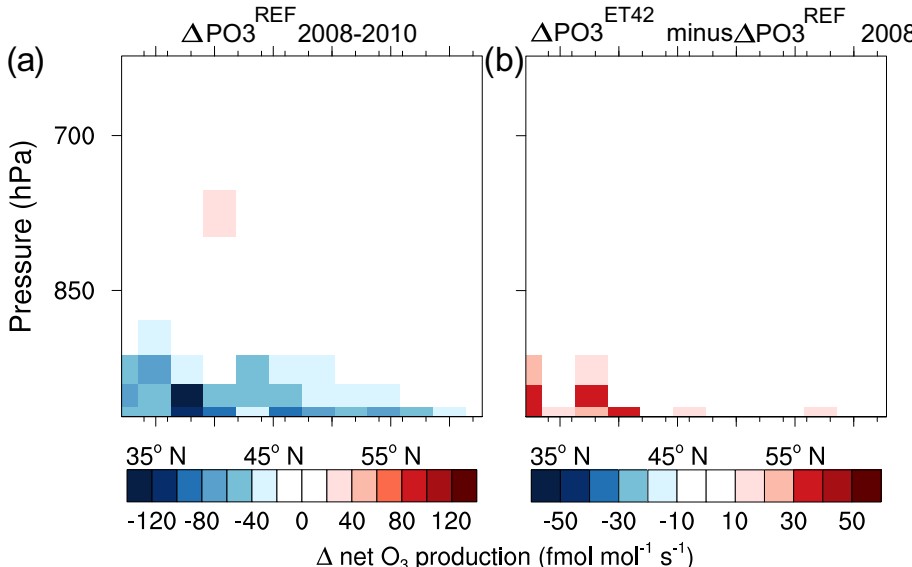

**Figure 6.** Zonally averaged differences of $P_{O3}$ ($\Delta P_{O3}$) between CM50 and EMAC (in $\mathrm{fmol\ mol^{-1}\ s^{-1}}$). **(a)** $\Delta P_{O3}$ calculated from the results of the *REF* simulation for JJA 2008–2010. **(b)** differences of $\Delta P_{O3}$ between the *ET42* and *REF* simulations for the year 2008 only. The CM50 data have been transformed on the horizontal and vertical grid of EMAC.

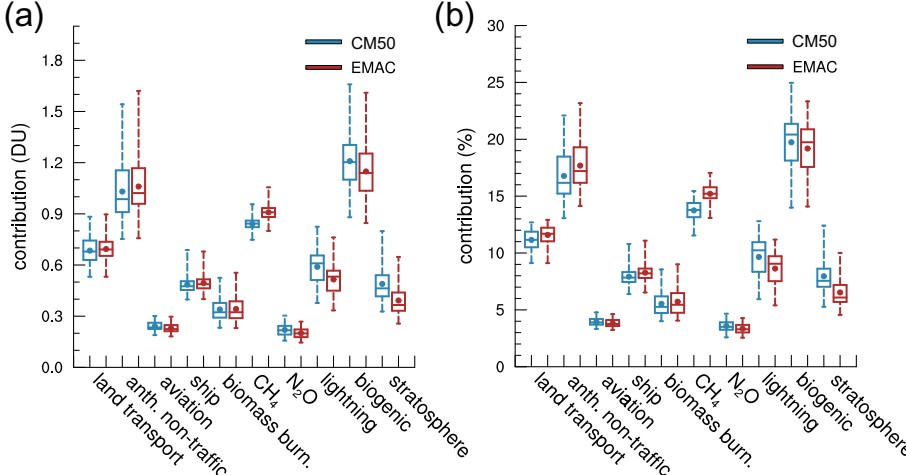

**Figure 7.** Box and whisker plot for the absolute (a, in DU) and relative (b, in %) contribution to the ozone column up to 850 hPa. The values are area-averaged over the CM50 domain. The lower and upper ends of the boxes indicates the 25th and 75th percentiles, the bars the medians, the dots the average and the whiskers the ranges of the timeseries for the JJA values from 2008–2010.

categories, both with contributions of more than 1 DU corresponding to more than 15 %. Both model instances simulate similar absolute ozone contributions of the categories anthropogenic non-traffic ($\approx$ 1.0 DU), land transport ($\approx$ 0.7 DU),

ship ($\approx$ 0.5 DU) and biomass burning ($\approx$ 0.4 DU). For the biogenic category, CM50 calculates slightly larger absolute contributions compared to EMAC (see Sect. 4.2), but the differences are small compared to the temporal variability of the contributions. Further, CM50 calculates larger absolute contributions of the categories lightning and stratosphere. This affects mainly the categories land transport, anthropogenic non-traffic, shipping and biomass burning, where EMAC simulates 0.1 to

5 around 1 percentage points larger relative contributions compared to CM50. At the same time the increased vertical mixing in CM50 leads to an increase of the relative contributions of the categories stratosphere, lightning and aviation compared to EMAC. Here, the differences are in the range of 0.1 to around 1.5 percentage points.

The positive ozone bias of CM50 compared to EMAC indicates a too efficient vertical mixing in CM50 (see Sect. 3). Therefore, the larger contributions of the categories stratosphere and lightning in CM50 compared to EMAC are likely an

10 artefact of this too efficient vertical mixing. However, partly this could be a feature of the increased resolution, as individual stratosphere-troposphere-exchange (STE) events are better represented in CM50 compared to EMAC due to the increased resolution (Hofmann et al., 2016; Mertens et al., 2016). Generally the correct representation of STE events poses a big challenge in most models (e.g. Zhang et al., 2011; Lin et al., 2012; Lefohn et al., 2014) and our results suggest a large difference of the contributin of STE to ground-level ozone between the results of different models.

The values which we discussed so far, however, are averages on continental scale. On the regional scale the differences can be much larger. Geographical distributions of the differences for the absolute and relative contributions as simulated by EMAC and CM50 are given in the Supplement (Fig. S3 and Fig. S4). Exemplarily, we want to focus on the categories land transport, as one important anthropogenic emission source, and biogenic emissions. As discussed in Sect. 2, the biogenic emissions are calculated on-line by both model instances depending on the meteorology and surface properties. While the total emissions

are comparable, the geographical distribution, as well as the area averaged contribution, differ (see Supplement Fig. S17 and Tables S2 to S10). As differences of on-line simulated emissions are a typical inter-model difference, a detailed investigation of the influence of these differences is of interest.

## 4.1 Contribution of land transport emissions to ground-level ozone

Averaged over JJA 2008 and the European area (defined as rectangular box from $10°$ W: $30°$ E and $32°$ N: $65°$ N, see red

square in Fig. 1) EMAC simulates a relative contribution of the land transport emissions (denoted as $O_3^{tra}$) to ground-level ozone of 13.1 %, while CM50 calculates a contribution of 11.9 %. A decrease of the emission resolution in CM50 increases the relative contribution to 12.1 % (*ET42* simulation), and the change of the anthropogenic emission inventory in CM50 increases the contribution to 12.7 % (*EVEU* simulation). In all cases similar absolute contributions of $O_3^{tra}$ are simulated which range between 6.0 and 6.4 $\mathrm{nmol\,mol^{-1}}$. The area averaged values indicate that the inter-model differences between

CM50 and EMAC as discussed in detail in Sect. 3 have a larger influence on the calculated contributions than the change of the anthropogenic emission inventory. The impact of the coarsely resolved emission inventory on the area averaged values is rather small. In general, the differences of the average contributions of $O_3^{tra}$ simulated by the two model instances (EMAC and CM50), as well as simulated by CM50 for the four different simulations are $\approx 10$ % at maximum. In comparison to this, the

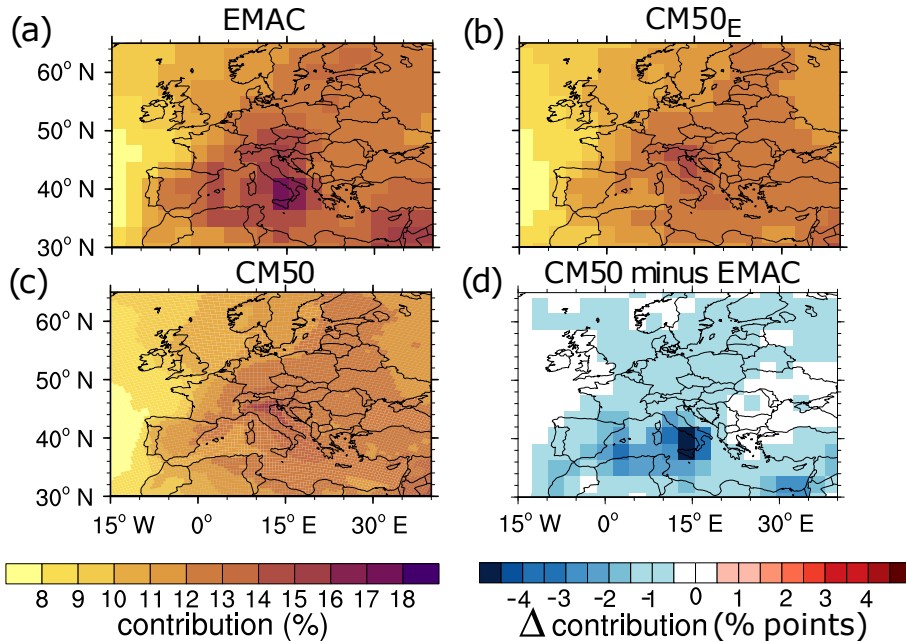

**Figure 8.** Comparison of the JJA averaged relative contribution of $O_3^{tra}$ to groundlevel $O_3$ (in %) of EMAC and CM50: **(a)** results of EMAC, **(b)** results of CM50 transformed onto the EMAC grid, **(c)** results of CM50 on the original grid and **(d)** difference ('CM50 minus EMAC' in percentage points) on the coarse grid. **(a)**– **(c)** use the same (left) colour bar. Shown are the results of the *REF* simulation, averaged for 2008–2010.

differences of the contributions to ground-level $O_3$ between EMAC and CM50 of the categories lightning and stratosphere are much larger ($\approx 20\ \%$ and $\approx 30\ \%$, respectively).

Regionally, the differences in relative contribution of $O_3^{tra}$ to ground-level ozone (see Fig. 8) can be larger than the area averaged differences. In general, both model instances simulate a comparable distribution with the largest relative contribution

of $O_3^{tra}$ in the Mediterranean region and contributions of around 8 % over the western Atlantic. These values are larger (10–18 %) over the continent. CM50 simulates a 0.5–1 percentage points lower relative contribution compared to EMAC. As discussed before, this is partly caused by to stronger vertical mixing and reduced ozone production ($P_{O3}$) in CM50 compared to EMAC. With increasing altitude the differences between EMAC and CM50 decrease (see Fig. S5 in the Supplement).

The largest differences of the relative contribution of $O_3^{tra}$ to ground-level ozone are simulated around the Mediterranean

area. The differences over the Mediterranean Sea (up to 2 percentage points and more, corresponding to more than 10 percent) can partly be attributed to the coarse resolution of the emissions in EMAC compared to CM50. The coarse resolution leads to an artificial increase of $P_{O3}$ (see Sect. 3.1) which in turn leads to an increase of the contribution from $O_3^{tra}$ (and other anthropogenic categories). Accordingly, the results of CM50 of the *ET42* simulation shows regionally up to 3 nmol mol$^{-1}$ and 3 percentage points larger contributions of land transport emissions to ozone as the *REF* simulation (see also Fig. S7 in the

Supplement). However, especially the large differences over Southern Italy and Sicily between CM50 and EMAC can not be

attributed to the coarse resolution of the emissions. Here, EMAC simulates the largest contribution (up to 17 %) in the European region (especially around the Naples region with large land transport emissions), while CM50 simulates contributions of around 13 %. On the coarse EMAC grid most parts of Southern Italy are considered as sea, affecting especially the calculation of dry deposition in EMAC, as dry deposition of ozone is lower over sea as over land. Therefore, the coarse resolution of the land sea

mask in EMAC compared to CM50 leads to an artificial underestimation of the ozone dry deposition in EMAC. In addition, the coarse land sea mask leads to differences in the calculation of biogenic emissions. Especially over Sicily, EMAC simulates no biogenic emissions (including soil-$NO_x$) while CM50 simulates large emissions here (see Fig. S17 in the Supplement). Accordingly, soil-$NO_x$ and anthropogenic $NO_x$ do not compete in EMAC in this area and ozone is mostly formed from anthropogenic emissions. Compared to these artificial peaks simulated by EMAC around Naples and over Sicily, CM50 shows

the largest contribution (up to 15 %) around the Po Valley. In this region, large amounts of emissions by land transport take place and ozone production is enhanced by stable and sunny weather conditions. The differences between EMAC and CM50 around the Naples region are even larger (up to 6 percentage points, see Fig. S6 in the Supplement) for the extreme values (95th percentile) as for the mean values which were discussed so far. Accordingly, extreme values are even stronger deteriorated as the mean values by the coarse land-sea mask problems discussed above.

The further increase of resolution from 50 km (CM50) to 12 km (CM12) impacts ozone and the contributions of ozone only slightly (see Fig. S11 in the Supplement). In general, we note a decrease of the absolute ozone values, as well as the absolute contributions of anthropogenic emissions (including the land transport category) near the hotspot regions (e.g. Rhine-Ruhr, Munich, and Frankfurt), if the model resolution is increased (*REF* simulation). The increase of the resolution of the emission inventory (*EVEU* simulation) intensifies this effect, i.e. near the hotspots ozone values and absolute contributions of $O_3^{\mathrm{tra}}$

decrease further. In Southern and Eastern Germany, however, the ozone values increase. As a comparison of the contributions of the individual tagging categories shows, this is mainly caused by an increase of the contribution from stratospheric ozone and from the $CH_4$ category. The increase of stratospheric ozone is partly caused by the enhanced topography in CM12 compared to CM50 as well as larger convective up- and downdraft massfluxes in CM12 compared to CM50 The larger contribution of ozone from the $CH_4$ category (meaning more ozone formed by reactions involving $CH_4$ oxidation products) is consistent with

the findings of the larger tropospheric oxidation capacity (i.e. lower methane lifetime) in CM12 compared to CM50 by Mertens et al. (2016).

    CM12 simulates a lower relative contribution of $O_3^{\mathrm{tra}}$ to ground-level $O_3$ over Germany than CM50 (see Fig. 9). The difference is largest in Southern Germany, however, mostly below 0.5 percentage points (corresponding to less than 5 %). The differences of the mean and 95th percentile (see Fig. S12 in the Supplement) of the contributions of $O_3^{\mathrm{tra}}$ between CM12

and CM50 are much smaller compared to the differences caused by the different anthropogenic emissions inventory (e.g. the differences of the results of the *REF* and *EVEU* simulation). Accordingly, the differences of emission inventories dominate over differences caused by the resolution of emission inventories and models when comparing the results of CM50 and CM12.

    What is not discussed here in detail is the influence of the difference of the shorter lived species, e.g. $NO_2$ or the tagged contributions to $NO_y$, which largely differ between the two resolutions. Here, maxima (e.g. in Stuttgart or around the Rhine-

Ruhr area) are displaced in the coarser resolution (CM50) compared to the finer resolution (CM12). However, the direct

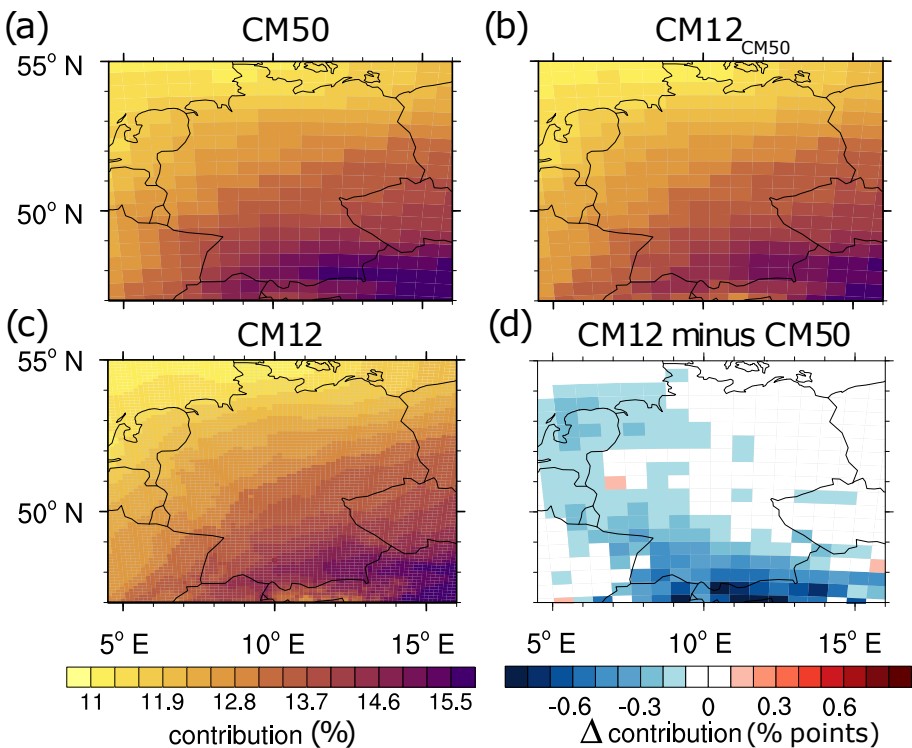

**Figure 9.** Comparison of the JJA averaged ground-level contribution of $O_3^{tra}$ to $O_3$ (in %) of CM50 and CM12: **(a)** results of CM50, **(b)** results of CM12 transformed onto the CM50 grid, **(c)** results of CM12 on the original grid and **(d)** difference ('CM12 minus CM50' in percentage points) on the coarse grid. **(a)**–**(c)** use the same (left) colour bar. Shown are the results of the *EVEU* simulation, averaged for 2008.

influence of displaced precursors on ozone itself is not very large, because ozone formation usually takes place downwind of the source itself. Further, compared to previous studies investigating the influence of the model/emission inventory resolution on ozone (e.g. Wild, 2007; Tie et al., 2010; Markakis et al., 2015), it is important to note that we apply a chemistry-climate model in which not only the chemical processes are calculated on the finer grid, but also the meteorology. This can alter the results compared to studies applying simpler chemistry-transport models.

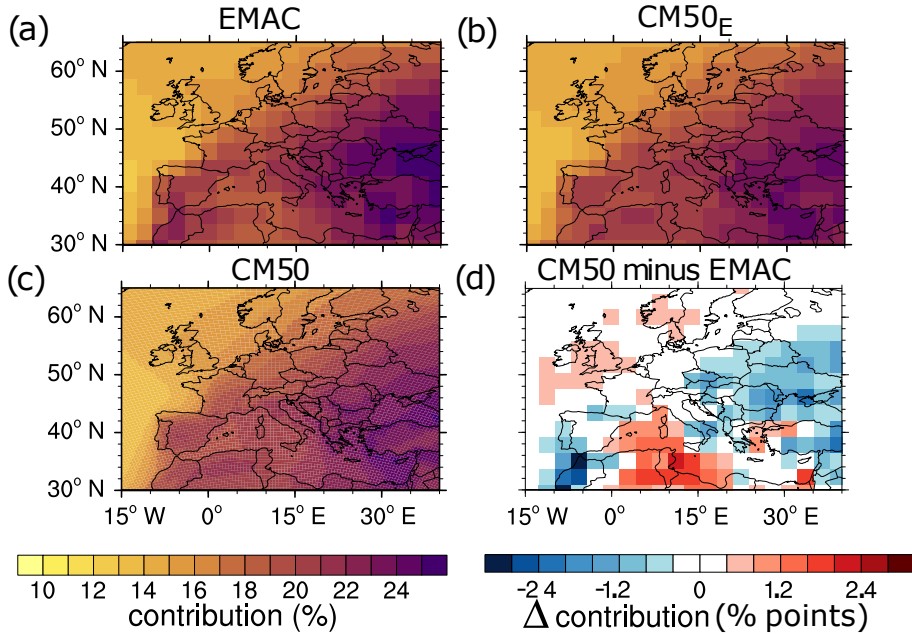

**Figure 10.** Comparison of the JJA averaged ground-level contribution of $O_3^{soi}$ to $O_3$ (in %) of EMAC and COSMO-CLM/MESSy: **(a)** results of EMAC, **(b)** results of CM50 transformed onto the EMAC grid, **(c)** results of CM50 on the original grid and **(d)** difference ('CM50 minus EMAC' in percentage points) on the coarse grid. **(a)**– **(c)** use the same (left) colour bar. Shown are the results of the *REF* simulation, averaged for 2008–2010.

## 4.2 Contribution of biogenic emissions to ground-level ozone

The JJA 2008 averaged relative contribution of ozone from biogenic emissions (mainly soil-$NO_x$ and biogenic $C_5H_8$, denoted as $O_3^{soi}$) to ground-level $O_3$ in Europe (see Sect. 4.1 for the definition) range from 19.0 to 19.6 % in all simulations. Hence, the differences of the relative contribution of $O_3^{soi}$ to ground-level ozone on the continental scale are rather small (below 5 %).

5 The same is true for the absolute values, ranging from 9.3 to 9.7 $\mathrm{nmol\ mol^{-1}}$.

With respect to the geographical distribution (Fig. 10) EMAC and CM50 simulate a strong North-West to South-East gradient with relative contributions from $O_3^{soi}$ of around 10 % over the Atlantic and more than 20 % over South-Eastern Europe. In contrast to the contribution of $O_3^{tra}$, EMAC simulates not generally larger contributions of $O_3^{soi}$ as CM50. Instead, EMAC simulates (*REF* simulation) larger contributions (1–2 percentage points) over South-Eastern Europe and Morocco/Iberian Peninsula,

10 while CM50 simulates around 1–2 percentage points larger contributions over large parts of the Mediterranean Sea as well as over Northern Africa. Also around the British Islands and Norway, CM50 simulates around 0.5 percentage points larger contributions of $O_3^{soi}$ than EMAC. Averaged over then CM50 domain, CM50 ends up with 0.5 percentage points larger contributions of $O_3^{soi}$ than EMAC. Similar as for the land transport category, the differences between the results of both model instances decrease with increasing height, but the general pattern stays similarly (see Fig. S8 in the Supplement).

The differences between EMAC and CM50 are only partly caused by the different geographical distribution of the biogenic emissions in EMAC compared to CM50. When applying the biogenic emissions as calculated by EMAC in CM50 (*EBIO* simulation) the relative and absolute contributions of $O_3^{soi}$ increase mainly in the Mediterranean area by up to 2 percentage points and 3 nmol mol$^{-1}$, respectively (see Fig. S9 and Fig. S10 in the Supplement). The characteristic dipole pattern with lower contributions of $O_3^{soi}$ in South-Eastern Europe and larger contributions in Southern Europe and Northern Africa in CM50 compared to EMAC remains similar. This pattern can partly be attributed to the coarse resolution of the shipping emissions in EMAC, leading to a positive ozone bias in the Mediterranean sea (see Sect. 3). The dipole pattern, however is neither caused by the coarse resolution of the emissions nor by different biogenic emissions, but mainly caused by the differences of the meteorology simulated by EMAC and CM50.

In general, we conclude that regionally differences of the relative and absolute contribution of $O_3^{soi}$ caused by inter-model differences, emission resolution as well as different geographical distribution are up to 15 %. Averaged over Europe the differences are lower (10 %). Again, the differences are lower as for example the differences of around 30 % observed for the differences of the contributions to ozone from the stratosphere.

## 5  Discussion

So far, the results indicate that with respect to average values on continental scale, the differences caused by the resolutions of the model/emission inventory are rather small. This confirms findings by Stock et al. (2013), reporting only a small influence of the global redistribution of megacity emissions (which can be seen as a locally decreased emission resolution) on the global ozone budget.

To summarise and quantify these differences in more detail, Fig. 11 shows the absolute (a) and relative (b) contributions of $O_3^{tra}$ to ground-level ozone averaged over the CM50 domain, as well as for the geographical regions defined in the PRUDENCE project (Christensen et al., 2007). The results of EMAC are not analysed for these geographical regions, as due to the coarse resolution some regions would only consist of a few grid points.

Figure 11 shows that also on the scale of smaller regions, the absolute and the relative contribution of $O_3^{tra}$ to ground-level ozone is only slightly influenced by the coarse resolution of anthropogenic emission inventories (*ET42*) as well as by a different geographical location and resolution of biogenic emissions (*EBIO*). This does not only hold for the mean $O_3^{tra}$ contributions, but also for the extreme values expressed by the 95th percentile. Further, also the simulated differences for the biogenic and shipping category, which are more affected by the differences of the emission inventories in the two simulations, are rather small (see Fig. S13 and Fig. S14 in the Supplement). The largest simulated differences of the mean contribution of shipping emissions to ground-level ozone between the *REF*, *EBIO* and *ET42* simulation are around 0.5 nmol mol$^{-1}$ and below 0.5 percentage points, respectively. The largest change (95th percentile) of the biogenic category in the region Iberian Peninsula is around 0.7 nmol mol$^{-1}$ and 0.5 percentage points.

Compared to the differences of the contribution of $O_3^{tra}$ between the *REF*, *ET42*, and *EBIO*, the differences caused by a changed emission inventory (*EVEU*) are larger. In the Mediterranean region, the mean and 95th percentile of the contribution

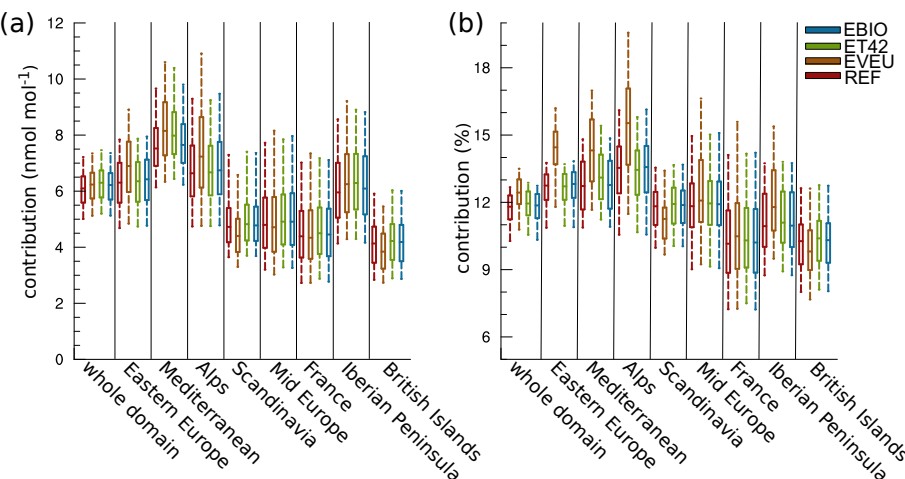

**Figure 11.** Comparison of the contributions of $O_3^{tra}$ to ground-level ozone for JJA 2008 between the four simulations. (a) displays the absolute contribution in $nmol\,mol^{-1}$ and (b) the relative contribution to ground-level ozone (in %). All values are area averaged over the respective region and are calculated using the results of the CM50 instance. The lower and upper end of the box indicate the 25th and 75th percentile, respectively, the bar the median, and the whiskers the 5th and 95th percentile of the timeseries for the JJA values from 2008 based on 3-hourly model output.

of $O_3^{tra}$ increases by 1 $nmol\,mol^{-1}$ and 2 percentage points, respectively. In the Alps region the increase of the 95th percentile of the contribution is up to 1.3 $nmol\,mol^{-1}$ and 3 percentage points, respectively. Similarly, also for the contribution of shipping emissions the differences are largest with the changed emission inventory (up to 1.5 $nmol\,mol^{-1}$ and 1 percentage point). Accordingly, changes in the resolution of the emission inventory or the biogenic emissions can affect the contribution of

5 anthropogenic categories (such as land transport and shipping). However, on the regional scale the main drivers of uncertainties are clearly the anthropogenic emissions and and differences caused by the model resolution and/or model differences. As an example we found regional differences (cf. Sect. 4.1) of the contribution of $O_3^{tra}$ to ground-level $O_3$ between EMAC and CM50 of up to 20 % around the Naples region, which in this case can mainly be attributed to the coarse land-sea mask of EMAC, leading to the emission of land transport emissions over the sea.

The results of the model evaluation, however, are not very helpful in judging which of the two emission inventories are more realistic. Although, *EVEU* shows a smaller ozone bias compared to *REF* caused by reduced precursor emissions, it is unclear if lower anthropogenic non-traffic emissions in the VEU compared to MAC emission inventories is realistic.

## 6   Summary and Conclusions

In the present study, we are focusing on the question "Are contributions of emissions to ozone a matter of scale?". To answer this
question we compare the influence of the model, the model resolution, the emission resolution and the emission inventory on the results of ozone contribution analyses. For this we apply the MECO(n) model system which combines a global and a regional

model by means of an on-line nesting technique. By applying the identical tagging diagnostics (source apportionment method) in the regional and global model and consistent boundary conditions, we are able to compare the results of model instances with different resolutions to investigate the influence of the model and emission inventory resolutions onto the diagnosed ozone contributions. Such analyses are important for quantifying uncertainties of ozone source apportionment studies, which arise due to limitations of the model and/or computational resources.

For the specific model set-up involving the global model EMAC and the regional model COSMO-CLM/MESSy our results show that simulated differences of ozone contributions on continental scale (e.g. Europe) are rather small. The largest differences of the contribution of anthropogenic emission sources was up to 10 % for the contribution of land transport emissions to ground-level ozone. However, the contribution of stratospheric ozone to ground-level ozone calculated by EMAC and COSMO differs by up to 30 %. One main reason for this large difference of the contributions of stratospheric ozone between the two models are the enhanced vertical mixing and larger convective up- and downdrafts in COSMO-CLM/MESSy than in EMAC. Taking the comparison with the measurements into account the vertical mixing in COSMO-CLM/MESSy and the enhanced stratospheric contribution are likely too large. On the regional scale the differences between the contributions of anthropogenic emission sources simulated by COSMO-CLM/MESSy and EMAC are much larger. Here, we observed differences of up to 20 % for the contributions of land transport emissions to ground-level ozone. This difference is mainly caused by the coarse land-sea mask of the global model instance, leading to emissions of land transport emissions over sea, different ozone dry deposition and missing biogenic emissions. Taking the results of the same model instance (CM50) into account the largest influence on the results are caused by different emissions inventories. However, locally also coarse resolved emission inventories and differences of the biogenic emissions can lead to differences of up to 20 %. In addition, we showed how the differences of the source apportionment results between different model instances can help to explain model biases and the physical/chemical mechanisms causing these biases.

Apart from many model specific findings of this study, its results have important implications for other modelling studies and modellers applying source apportionment methods. These implications are:

– First, our study shows that average continental contributions of anthropogenic emissions are quite robust with respect to the used model and the used model resolution. This means that global models at coarse resolution can be used to perform ozone source apportionment in the global context.

– Second, our results also show that on the regional scale, the differences either caused by different models, but also by model resolution can be larger. These effects arise mainly near hotspot regions like the Po Valley or near major shipping routes in the Mediterranean Sea. However, especially in these areas, contribution analyses of anthropogenic emissions are very important and spurious effects, such as artificially increased ozone levels and contributions caused by the coarse resolution of models and or emission inventories should be avoided. Hence, for regional analyses fine resolved models and emission inventories are required.

– Third, our results clearly indicate how large the spread between models with respect to STE is. The importance of stratospheric ozone, both in the global and the regional model, corroborates the necessity of tracing the contributions of

stratospheric ozone to ground-level ozone explicitly by the source apportionment methods. However, only few currently available methods used on the regional scale account for this process.

Clearly, this study is only a first step to quantify the driving sources of uncertainties and especially the role of the model and emission inventory resolutions on the results of ozone contribution studies. Especially, as some processes like vertical diffusion or vertical transport can heavily alter the model results, follow up studies need to take into account more (and more different) models to better quantify the uncertainties due to differences of the meteorology simulated by different models. In addition, the two analysed anthropogenic emission inventories clearly do not reflect the whole spectrum of different emission estimates. Further, our analyses focused only on differences near the origin of the emissions. An increased resolution leads to a more realistic chemistry within the plumes downwind of the emission hotspots. This can affect the long range transport from different precursors and might influence regions far away from the emission region. Especially calculations of radiative forcings are very sensitive to ozone near the tropopause. In a coarsely resolved model, the overestimated absolute contributions might lead to a biased radiative forcing. This effect, however, is difficult to quantify and would require very fine resolved global chemistry climate models or 2-way-nesting capabilities, which feed back information about the contributions from the fine back to the coarse grid. For a next step a further increase of the model and emission resolution should be envisaged. Even if we found only small differences between 50 and 12 km resolution this step would be important, as even with a 12 km grid resolution emissions are diluted over large areas. A finer resolution could reduce the dilution strongly. Such an analysis, however, is hindered by two aspects: First, consistent emission inventories (anthropogenic and natural) with a resolution of 1 km over areas, which are large enough to compare models on regional and global scale must be available. Second, requirements with respect to computational time of chemistry-climate models with $\approx$ 1 km resolution over large computational domains are very demanding, hindering detailed quantification of the differences caused by the resolution over long integration periods.

*Code and data availability.* The Modular Earth Submodel System (MESSy) is continuously further developed and applied by a consortium of institutions. The usage of MESSy and access to the source code is licenced to all affiliates of institutions which are members of the MESSy Consortium. Institutions can become a member of the MESSy Consortium by signing the MESSy Memorandum of Understanding. More information, including on how to become licensee for the required third party software, can be found on the MESSy Consortium Website (http://www.messy-interface.org). The code presented here has been based on MESSy version 2.50 and is available in the official release (version 2.51). The namelist set-up used for the simulations is part of the electronic supplement. The data used for the figures 6 to 11 are part of the electronic supplement.

**Appendix A: Definition of RMSE and MB**

We define the root mean square error (RMSE) as:

$$RMSE = \sqrt{\frac{1}{n}\Sigma_{i=1}^{n}\left(\mathrm{O}_{3_i}^{mod} - \mathrm{O}_{3_i}^{meas}\right)^2}, \tag{A1}$$

where $n$ is the number of data points, $O_3^{mod}$ the simulated and $O_3^{meas}$ the measured ozone concentrations.

The normalized mean bias error (MB) is defined as:

$$MB = \left(\frac{\overline{O_3^{mod}}}{\overline{O_3^{meas}}} - 1\right) \cdot 100, \tag{A2}$$

where $\overline{O_3^{mod}}$ and $\overline{O_3^{meas}}$ are the simulated and measured ozone concentrations averaged for all stations and month, respec-

5    tively.

*Competing interests.* The authors declare that they have no competing interests.

*Acknowledgements.* M. Mertens acknowledges funding by the DLR projects 'Verkehr in Europa' and 'Auswirkungen von $NO_x$'. Furthermore, part of this work is funded by the DLR project 'VEU2'. A. Kerkweg acknowledges funding by the German Ministry of Education and Research (BMBF) in the framework of the MiKlip (Mittelfristige Klimaprognose/Decadal Prediction) subproject FLAGSHIP (Feedback of a

10   Limited-Area model to the Global-Scale implemented for HIndcasts and Projections, funding ID 01LP1127A). We thank R. Eichinger (DLR) and M. Kilian (DLR) for very valuable comments improving the manuscript considerably. Further, we acknowledge the comments from three anonymous referees, which improved the manuscript. We acknowledge the Leibniz-Rechenzentrum in Garching for providing computational resources on the SuperMUC2 under the project id PR94RI. Analysis and graphics for the data used were performed using the NCAR Command Language (version 6.4.0) software developed by UCAR/NCAR/CISL/TDD and available online: https://doi.org/10.5065/D6WD3XH5

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
