# Peer review of "Are contributions of emissions to ozone a matter of scale? - A study using MECO(n) (MESSy v2.50)"

_Geoscientific Model Development, 2019_

## Referee Comment (RC1) · Anonymous Referee #1 · 15 Jun 2019

This paper presents an analysis of simulations at various horizontal resolutions, with emissions at various resolutions, and different emissions inventories.

The research is technically sound, and the application of source tagging and attribution is well illustrated. However, the paper does not seem to have any new results. The models and tagging technique used have all been published previously. The majority of their conclusions confirm previous work. Their strongest conclusion seems to be that different emissions inventories making the largest difference in ozone simulations, which I think is well known, but they do not offer any assessment about which might be more accurate. If the authors feel they have more compelling results, then they should

make them much clearer.

It is not apparent why the authors thought GMD was the best journal for this work. It does not seem to have any new model development, or even quantitative evaluation of the model. The paper reads very much like a technical report for MESSy users. For example, it would help the general reader if 'ONEMIS' was defined and explained on p.5.

While I see no errors in this work, I feel significant revisions are required to make it suitable for publication. The paper should emphasize new results, not the confirmation of previous results. It would also be valuable to include comparisons to observations, and perhaps then conclusions can be drawn as to how fine does model horizontal resolution need to be to reproduce observations, and to reproduce accurately physical phenomena (e.g., vertical transport) that affect ozone distributions.

Technical corrections

p.4, l.17: 'to calculate' should be 'calculation of'

p.4, l.31 and elsewhere: 'lighting' should be 'lightning'

p.6, l.17: See -> Sea

p.14, l.27+: use "%" instead of "percentage points"; also 'respectively' is unnecessary.

p.15, l.1: 'effect' -> 'affect'

p.15, l.12: 'to quantify' -> 'for quantifying'

---

## Author Comment (AC1) · 21 Jun 2019

Dear reviewer#1,

thank you very much for your review of our manuscript GMD-2019-07. We like to post a short comment to your review, as we see some points you raised differently. After the discussion phase has ended we will provide a full answer. In the following, referee comments are given in italics, our reply's in normal font.

*This paper presents an analysis of simulations at various horizontal resolutions, with*

*emissions at various resolutions, and different emissions inventories.*

Reply: To be more precise our analysis focuses only on diagnosed ozone contributions and uncertainties of these contributions which arise due to model limitations (e.g. resolution, parametrisation), limited resolution of emission inventories and uncertainties of the emission inventories.

*The research is technically sound, and the application of source tagging and attribution is well illustrated. However, the paper does not seem to have any new results. The models and tagging technique used have all been published previously. The majority of their conclusions confirm previous work. Their strongest conclusion seems to be that different emissions inventories making the largest difference in ozone simulations, which I think is well known, but they do not offer any assessment about which might be more accurate. If the authors feel they have more compelling results, then they should make them much clearer.*

Reply: First of all thank you very much for honouring our work. Indeed our analysis is very technical and focuses on the impact of technical limitations of models on the results of source apportionment diagnostics. However, we do not agree with reviewer#1 that our manuscript does not show any new results. Clearly, the dependence of simulated ozone concentrations on the resolutions of model and emissions are well known (see p1l4f, p2l8ff of our manuscript), and where appropriate we cite previous literature. The focus of our manuscript, however, is not on simulated ozone concentrations but on diagnosed contributions to ozone. We are not aware of any previous publication, which investigates the impact of these factors on the results of a source apportionment (e.g. tagging) method. Further, we are not aware of any similar model system allowing for such an analysis, as it requires a consistent global-regional model chain applying the identical source-attribution method on the global and regional scale. Previous publications applying source attribution on the regional scale (e.g. Dunker et al, 2002, Li et

al., 2012, Kwok et al., 2015, Valverde et al., 2016, Karamchandani et al., 2017) considered only the contributions as simulated by the regional model and cannot attribute ozone transported from the stratosphere or across the regional model domain to specific emission categories. In the revised manuscript, we will clarify the focus and the new insights accordingly.

Finally, we would like to remark that publications in GMD are not about presenting new scientific results. Publications in GMD are mainly to document model developments, document experimental set-ups of model simulations, document evaluation of model systems, present model evaluation strategies and to present technical analyses of model systems. We think that our manuscript documents the influence of model and emission inventory resolutions on source attribution results. This is clearly important to asses source apportionment results and there related uncertainties, also for other model systems.

*It is not apparent why the authors thought GMD was the best journal for this work. It does not seem to have any new model development, or even quantitative evaluation of the model.*

Reply: The main goal of our manuscript is to analyse the impact of technical differences (e.g. model and emission inventory resolutions and/or the applied model) on the simulated contributions to ozone. Accordingly, our research questions are rather technical and focus on the impact of differences due to model limitations and/or differences due to input data. This does not necessarily result in new scientific results, but it is certainly important for other researchers in the same field. Therefore, we chose GMD instead of ACP as journal and choose "development and technical paper" as manuscript type. These type of manuscripts also includes: '[...] papers relating to technical aspects of running models and the reproducibility of results' (GMD website).

*The paper reads very much like a technical report for MESSy users. For example, it would help the general reader if 'ONEMIS' was defined and explained on p.5.*

Reply: Of course the paper should not read as a technical report to MESSy users. Even tough the specific results we discuss are only valid for the specific model system and set-up (as it is common for most model studies) the general conclusions (see next paragraph) are also important to other researchers using source apportionment methods in a variety of models (e.g. CMAQ, WRF). As discussed in the next paragraph, we will revise the manuscript in such a way that the general findings, which are important for the whole community, will become more clear.

For the revised manuscript we will carefully check the manuscript again and describe specifics of the MESSy world, which are not defined in detail. For your specific example of ONEMIS on p5l1f we write: 'Emissions of soil-NOx and biogenic isoprene ($C_5H_8$) are calculated by the MESSy submodel ONEMIS (Kerkweg et al, 2006) , which uses the parametrisations of Yienger and Levy (1995) for soil-NOx, and Guenther et al. (1995) for $C_5H_8$.' To our understanding this is a definition of ONEMIS. We could of course describe the processes in ONEMIS in more detail, but this would lead in large parts to a reproduction of Kerkweg et al. (2006).

*While I see no errors in this work, I feel significant revisions are required to make it suitable for publication. The paper should emphasize new results, not the confirmation of previous results. It would also be valuable to include comparisons to observations, and perhaps then conclusions can be drawn as to how fine does model horizontal resolution need to be to reproduce observations, and to reproduce accurately physical phenomena (e.g., vertical transport) that affect ozone distributions.*

Reply: First of all we would like to thank reviewer#1 that she/he generally confirms

that our analysis does not have any errors. As mentioned above, we think that our study offers important new results, which are also important for communities outside the MESSy community. In particular, our research offers insights into uncertainties of diagnosed ozone contributions. These new results are:

- Diagnosed contributions of anthropogenic emissions are rather robust on the continental scale. Differences due to model, model and emission inventory resolutions and anthropogenic emissions are 10 % at maximum.

- Uncertainties of contributions at ground level due to downward transport of ozone are rather large. We find differences of up to 30 % on the continental scale.

- On the regional scale differences in contributions of land transport emissions are rather large and can reach up to 20 % and more, due to different reasons. Therefore high resolved model and high resolved emissions are important for regional assessments of ozone source apportionment.

- Source attribution diagnostics are a valuable tool to better understand inter-model differences.

However, the comment from reviewer#1 also clearly shows that we did not clearly bring up these new results. Therefore, we will highlight them more prominent in the revised manuscript.

We agree with reviewer#1 that a detailed comparison with observations is very valuable. However, ozone contributions cannot be measured directly. Therefore, more complex evaluation strategies involving proxies, which can be measured, are needed. However, this is beyond the scope of this manuscript, because we here focus on the influence of technical aspects and try to estimate uncertainties, which arise only due to technical limitations. In a follow up study we work on a more detailed

analysis involving detailed observations of specific measurement campaigns, which are confronted with modelled concentrations and diagnosed contributions to further constrain uncertainties of source attribution results.

Further, we are very thankful for the technical corrections provided by review#1 and will consider them for the revised version and answer them in our full reply.

We are looking forward to your reply,

Mariano Mertens
(on behalf of all co-authors)

**Bibliography**

Dunker, A. M., Yarwood, G., Ortmann, J. P., and Wilson, G. M.: Comparison of Source Apportionment and Source Sensitivity of Ozone in a Three-Dimensional Air Quality Model, Environmental Science Technology, 36, 29532964, doi:10.1021/es011418f, URL http://dx.doi.org/10.1021/es011418f, pMID: 12144273, 2002.

Guenther, A., Hewitt, C., E., D., Fall, R. G., C., Graedel, T., Harley, P., Klinger, L., Lerdau, M., McKay, W., Pierce, T., S., B., Steinbrecher,R., Tallamraju, R., Taylor, J., and Zimmermann, P.: A global model of natural volatile organic compound emissions, J. Geophys. Res., 15 100, 8873–8892, 1995.

Karamchandani, P., Long, Y., Pirovano, G., Balzarini, A., and Yarwood, G.: Source-sector contributions to European ozone and fine PM in 2010 using AQMEII modeling data, Atmospheric Chemistry and Physics, 17, 56435664, doi:10.5194/acp-17-5643-

2017, URL https://www.atmos-chem-phys.net/17/5643/2017/, 2017.

Kerkweg, A., Sander, R., Tost, H., and Jöckel, P.: Technical note: Implementation of prescribed (OFFLEM), calculated (ONLEM), and pseudo-emissions (TNUDGE) of chemical species in the Modular Earth Submodel System (MESSy), Atmos. Chem. Phys., 6, 36033609, doi:10.5194/acp-6-3603-2006, URL http://www.atmos-chem-phys.net/6/3603/2006/, 2006.

Kwok, R. H. F., Baker, K. R., Napelenok, S. L., and Tonnesen, G. S.: Photochemical grid model implementation and application of VOC, NOx, and O3 source apportionment, Geoscientific Model Development, 8, 99114, doi:10.5194/gmd-8-99-2015, URL http://www.geosci-model-dev.net/8/99/2015/, 2015.

Li, Y., Lau, A. K.-H., Fung, J. C.-H., Zheng, J. Y., Zhong, L. J., and Louie, P. K. K.: Ozone source apportionment (OSAT) to differentiate local regional and super-regional source contributions in the Pearl River Delta region, China, Journal of Geophysical Research: Atmospheres, 117, doi: 10.1029/2011JD017340, URL http://dx.doi.org/10.1029/2011JD017340, d15305, 2012.

Valverde, V., Pay, M. T., and Baldasano, J. M.: Ozone attributed to Madrid and Barcelona on-road transport emissions: Characterization of plume dynamics over the Iberian Peninsula, Science of The Total Environment, 543, Part A, 670 682, doi:http://dx.doi.org/10.1016/j.scitotenv.2015.11.070, URL http://www.sciencedirect.com/science/article/pii/S0048969715310500, 2016.

Yienger, J. J. and Levy, H.: Empirical model of global soil-biogenic NOx emissions, Journal of Geophysical Research: Atmospheres, 100, 11 44711 464, doi:

10.1029/95JD00370, URL http://dx.doi.org/10.1029/95JD00370, 1995

---

## Referee Comment (RC2) · Anonymous Referee #2 · 12 Jul 2019

This manuscript explores whether the source apportionment of surface ozone would be affected by model resolution. It performed model simulations with the different resolution of the model itself and the emission inventories. The difference in the source apportionment using a self-consistent tagging method is attributed to the model resolution and emission inventory resolution. The topic itself and the self-consistent tagging method are interesting. However, the analyses presented in the manuscript are too useful; the discussion and conclusions are not insightful (or not having any new results as pointed out by Anonymous Reviewer #1). I'd suggest the following items to improve the manuscript.

1. better defining the differences between simulations/models, be specific about what processes causing the variations in source apportionment. Here are just a few examples to improve.

(a) Meteorological inputs such as temperature and light are different in some simulations, which would result in different biogenic emissions in methods of 'on-line calculated' and 'calculated by EMAC.'

(b) Same anthropogenic emissions in different resolution might result in the same total emission but large regional differences. How do these emissions differ?

(c) What are actually causing the differences in STE flux in the coarse vs. fine resolution model? Could it be related to on-line vs. off-line meteorology/convections and/or temporal and horizontal averaging of meteorological inputs (just some examples I am familiar with, like in Yu et al. (2018) and Hu et al. (2017); certainly, many other literature on this topic are available)? The contribution from downward transport seems to be the largest differences among models, and it should be quite interesting to explore.

(d) It looks like the total lightning NOx emissions are the same across simulations, do their also have the same 3D distribution?

"Inter-model differences' should be better defined and documented and can provide insights on the calculated contributions. Specific discussion of these processes rather than vaguely saying because of the resolution would make this paper more useful.

2. the terms used in the manuscript are very confusing for readers from outside the MESSy model community, particularly when referring to the specific simulation. For example, CM50 is used to compare with EMAC, while one refers to the resolution of 50km of one model; the other refers to a different model. ET42 refers to "the MACCity emissions are transformed to the coarse grid of EMAC (T42), to investigate the impact of the resolution of the emission inventory.", but it sounds like it is done by the COSMO model only, so do all the REF, EBIO, EVEU simulations. Table 2 seems to suggest

that EMAC also has those four simulations. Table 1 is not useful in the context of this manuscript but just adds confusions by adding a bunch of acronyms. This manuscript should not be 'read very much like a technical report for MESSy users' as pointed by the other reviewer. Readability should be improved.

3. this paper could benefit from a section of model evaluation by adding comparisons with observations. This way could suggest which simulations are 'in practice' better and if the model simulations are actually realistic.

4. the metrics used to quantify simulation difference: this manuscript mostly uses the average concentration of ozone and relative contribution of a specific source. These tend only to show minimal differences among simulations; even though the manuscript claims 'up to 20%' in the calculated contribution of transport emissions, the absolute amount is small. One way to improve is looking at the probability distribution of concentrations or contributions, which could be much more useful to examine differences in model chemical pathways and for specific air pollution episodes, i.e., examples like in Fiore et al. (2002) and Yu et al. (2016).

Example literature

Fiore, A. M., Jacob, D. J., Bey, I., Yantosca, R. M., Field, B. D., Fusco, A. C., and Wilkinson, J. G., Background ozone over the United States in summer: Origin, trend, and contribution to pollution episodes, J. Geophys. Res., 107( D15), doi:10.1029/2001JD000982, 2002.

Yu, K., Jacob, D. J., Fisher, J. A., Kim, P. S., Marais, E. A., Miller, C. C., Travis, K. R., Zhu, L., Yantosca, R. M., Sulprizio, M. P., Cohen, R. C., Dibb, J. E., Fried, A., Mikoviny, T., Ryerson, T. B., Wennberg, P. O., and Wisthaler, A.: Sensitivity to grid resolution in the ability of a chemical transport model to simulate observed oxidant chemistry under high-isoprene conditions, Atmos. Chem. Phys., 16, 4369-4378, https://doi.org/10.5194/acp-16-4369-2016, 2016.

[Figure]

Hu, L., D. J. Jacob, X. Liu, Y. Zhang, L. Zhang, P. S. Kim, M. P. Sulprizio, and R. M. Yantosca (2017), Global budget of tropospheric ozone: Evaluating recent model advances with satellite (OMI), aircraft (IAGOS), and ozonesonde observations, Atmospheric Environment, 167, 323-334, doi:https://doi.org/10.1016/j.atmosenv.2017.08.036.

Yu, K., Keller, C. A., Jacob, D. J., Molod, A. M., Eastham, S. D., and Long, M. S.: Errors and improvements in the use of archived meteorological data for chemical transport modeling: an analysis using GEOS-Chem v11-01 driven by GEOS-5 meteorology, Geosci. Model Dev., 11, 305-319, https://doi.org/10.5194/gmd-11-305-2018, 2018.

---

## Author Comment (AC2) · 15 Sep 2019

Please find our full reply attached
* * *
[Figure]

Dear referee#1,

thank you very much for your review of our manuscript GMD-2019-07. After our short comment we would like to reply to your review in detail. In the following, referee comments are given in italics, our replies in normal font, and text passages which we included in the text are in bold.

*This paper presents an analysis of simulations at various horizontal resolutions, with emissions at various resolutions, and different emissions inventories.*

Reply: To be more precise, our analysis focuses on diagnosed ozone contributions and uncertainties of these contributions, which arise due to model limitations (e.g. resolution, parametrisations), limited resolution of emission inventories, and uncertainties of the emission inventories. To make this more clear we revised the manuscript at several points (see below) and add also an addition Section (Sect. 2.1) which discuss the source apportionment in more detail.

*The research is technically sound, and the application of source tagging and attribution is well illustrated. However, the paper does not seem to have any new results. The models and tagging technique used have all been published previously. The majority of their conclusions confirm previous work. Their strongest conclusion seems to be that different emissions inventories making the largest difference in ozone simulations, which I think is well known, but they do not offer any assessment about which might be more accurate. If the authors feel they have more compelling results, then they should make them much clearer.*

Reply: First of all thank you very much for honouring our work. Indeed our analysis is very technical and focuses on the impact of technical limitations of models on the results of source apportionment diagnostics. However, we do not agree with referee#2 that our manuscript does not show any new results. Clearly, the dependence of simulated ozone concentrations on the resolutions of model and emissions are well known (see p1l4f, p2l8ff of our manuscript), and where appropriate we cite previous literature. The focus of our manuscript, however, is not on simulated ozone concentrations but on diagnosed contributions to ozone. We are not aware of any previous publication, which investigates the impact of these factors on the results of a source apportionment (e.g. tagging) method. Further, we are not aware of any similar model system allowing for such an analysis, as it requires a consistent global-regional model chain applying the identical source-attribution method on the global and regional scale. Previous publications applying source attribution on the regional scale (e.g. Dunker et al., 2002; Li et al., 2012; Kwok et al., 2015; Valverde et al., 2016; Karamchandani et al., 2017) considered only the contributions as simulated by the regional model and are not able to attribute ozone transported from the stratosphere or across the lateral borders of the regional model domain to specific emission categories.

In addition, we would like to remark that publications in GMD are not primarily about presenting new scientific results. Publications in GMD are mainly to document model developments, document experimental set-ups of model simula-

**Fig. 1.**

---

## Author Comment (AC3) · 15 Sep 2019

Dear referee#1,
thank you very much for your review of our manuscript GMD-2019-07. After our short comment we would like to reply to your review in detail. In the following, referee comments are given in italics, our replies in normal font, and text passages which we included in the text are in bold.

*This paper presents an analysis of simulations at various horizontal resolutions, with emissions at various resolutions, and different emissions inventories.*
Reply: To be more precise, our analysis focuses on diagnosed ozone contributions and uncertainties of these contributions, which arise due to model limitations (e.g. resolution, parametrisations), limited resolution of emission inventories, and uncertainties of the emission inventories. To make this more clear we revised the manuscript at several points (see below) and add also an addition Section (Sect. 2.1) which discuss the source apportionment in more detail.

*The research is technically sound, and the application of source tagging and attribution is well illustrated. However, the paper does not seem to have any new results. The models and tagging technique used have all been published previously. The majority of their conclusions confirm previous work. Their strongest conclusion seems to be that different emissions inventories making the largest difference in ozone simulations, which I think is well known, but they do not offer any assessment about which might be more accurate. If the authors feel they have more compelling results, then they should make them much clearer.*

Reply: First of all thank you very much for honouring our work. Indeed our analysis is very technical and focuses on the impact of technical limitations of models on the results of source apportionment diagnostics. However, we do not agree with referee#2 that our manuscript does not show any new results. Clearly, the dependence of simulated ozone concentrations on the resolutions of model and emissions are well known (see p1l4f, p2l8ff of our manuscript), and where appropriate we cite previous literature. The focus of our manuscript, however, is not on simulated ozone concentrations but on diagnosed contributions to ozone. We are not aware of any previous publication, which investigates the impact of these factors on the results of a source apportionment (e.g. tagging) method. Further, we are not aware of any similar model system allowing for such an analysis, as it requires a consistent global-regional model chain applying the identical source-attribution method on the global and regional scale. Previous publications applying source attribution on the regional scale (e.g. Dunker et al., 2002; Li et al., 2012; Kwok et al., 2015; Valverde et al., 2016; Karamchandani et al., 2017) considered only the contributions as simulated by the regional model and are not able to attribute ozone transported from the stratosphere or across the lateral borders of the regional model domain to specific emission categories.

In addition, we would like to remark that publications in GMD are not primarily about presenting new scientific results. Publications in GMD are mainly to document model developments, document experimental set-ups of model simulations, document evaluation of model systems, present model evaluation strategies and to present technical analyses of model systems. We think that our manuscript documents the influence of model and emission inventory resolutions on source attribution results. This is clearly important to asses source apportionment results and their related uncertainties, also for other model systems.

To make the importance of our study also for other modelling communities more clear, we revised especially the conclusion (and the abstract) as discussed below in more detail.

*It is not apparent why the authors thought GMD was the best journal for this work. It does not seem to have any new model development, or even quantitative evaluation of the model.*

Reply: The main goal of our manuscript is to analyse the impact of technical limitations (e.g. model and emission inventory resolutions and/or the applied model) on the simulated contributions to ozone. Accordingly, our research questions are rather technical and focus on the impact of differences due to model limitations and/or differences due to input data. This does not necessarily imply in new scientific results, but it yields certainly important new insights for other researchers in the same field. Therefore, we chose GMD instead of ACP as journal and choose "development and technical paper" as manuscript type. These type of manuscripts also includes: '[...] papers relating to technical aspects of running models and the reproducibility of results' (GMD website).

*The paper reads very much like a technical report for MESSy users. For example, it would help the general reader if 'ONEMIS' was defined and explained on p.5.*

Reply: Of course the paper should not read as a technical report to MESSy users. Even tough the specific results we discuss are only valid for the specific model system and set-up (as it is common for most model studies) the general conclusions (see next paragraph) are also important to other researchers using source apportionment methods in a variety of models (e.g. CMAQ, WRF). As discussed in the next paragraph, we revised the manuscript in such a way that the general findings, which are important for the whole community, will become more clear.
For the revised manuscript we have carefully checked the manuscript again and describe specifics of the MESSy world, which are not defined in detail. For your specific example of ONEMIS on p5l1f we write: 'Emissions of soil-NOx and biogenic isoprene (C5H8) are calculated by the MESSy submodel ONEMIS (Kerkweg et al, 2006), which uses the parametrisations of Yienger and Levy (1995) for soil-NOx, and Guenther et al. (1995) for C5H8.'

*While I see no errors in this work, I feel significant revisions are required to*

*make it suitable for publication. The paper should emphasize new results, not the confirmation of previous results. It would also be valuable to include comparisons to observations, and perhaps then conclusions can be drawn as to how fine does model horizontal resolution need to be to reproduce observations, and to reproduce accurately physical phenomena (e.g., vertical transport) that affect ozone distributions.*

Reply: First of all we would like to thank referee#1 that she/he generally confirms that our analysis does not have any errors. As mentioned above, we think that our study offers important new results, which are also important for communities outside the MESSy community. In particular, our research offers insights into uncertainties of diagnosed ozone contributions. These new results are:

- Diagnosed contributions of anthropogenic emissions are rather robust on the continental scale. Differences due to the applied model, model and emission inventory resolutions and anthropogenic emissions are 10 % at maximum.

- Uncertainties of contributions at ground level due to downward transport of ozone are rather large. We find differences of up to 30 % on the continental scale.

- On the regional scale differences in contributions of land transport emissions are rather large and can reach up to 20 % and more, due to different reasons. Therefore fine resolved models and fine resolved emission inventories are important for regional assessments of ozone source apportionment.

- Source attribution diagnostics are a valuable tool to better understand inter-model differences.

However, the comment from referee#1 also clearly shows that we did not clearly bring up these new results. Therefore,we have highlighted the most important findings in more detail in our conclusion (and the abstract). The changed conclusion reads:

**Apart from many model specific findings of this study, its results have important implications for other modelling studies and modellers applying source apportionment methods. These implications are:**

- **First, our study shows that average continental contributions of anthropogenic emissions are quite robust with respect to the used model and the used model resolution. This means that global models at coarse resolution can be used to perform ozone source apportionment in this global context.**

- **Second, our results also show that on the regional scale, the differences either caused by different models, but also by model resolution are much larger. These effects arise mainly near hotspot**

**regions like the Po Valley or near major shipping routes in the Mediterranean Sea. However, especially in these areas, contribution analyses of anthropogenic emissions are very important and spurious effects, such as artificially increased ozone levels and contributions caused by the coarse resolution of models and or emission inventories should be avoided. Hence, for regional analyses fine resolved models and emission inventories are required.**

- **Third, our results clearly indicate how large the spread between models with respect to STE is. The importance of stratospheric ozone, both in the global and the regional model, corroborates the necessity of tracing the contributions of stratospheric ozone to ground level ozone explicitly by the source apportionment methods. However, only few currently available methods used on the regional scale account for this process.**

Further, we agree with referee#1 that a detailed comparison with observations is very valuable. However, ozone contributions cannot be measured directly. Therefore, more complex evaluation strategies involving proxies, which can be measured, are needed. This, however, is beyond the scope of this manuscript, because we here focus on the influence of technical aspects and try to estimate uncertainties, which arise only due to technical limitations. In a follow up study we work on a more detailed analysis involving detailed observations of specific measurement campaigns, which are confronted with simulated mixing ratios and diagnosed contributions to further constrain uncertainties of source attribution results. However, as also referee#2 asked for a section on model evaluation we added Sect. 3, with a basic model evaluation section focusing on ozone. This evaluation clearly indicates, that the vertical mixing of CM50 is too strong and CM50 likely overestimates the contributions of stratospheric ozone at the surface.

*p.4, l.17: 'to calculate' should be 'calculation of'*
Fixed. Thanks!
*p.4, l.31 and elsewhere: 'lighting' should be 'lightning'*
Indeed. Thanks!
*p.6, l.17: See -¿ Sea*
Fixed. Thanks!
*p.14, l.27+: use "%" instead of "percentage points"; also 'respectively' is unnecessary.*
We removed the respectively, but we stay with the percentage points. The difference in percentage points are obvious from the figure. Calculating % from the percentage-points might lead to missunderstandings.
*p.15, l.1: 'effect' -¿ 'affect'*
Changed
*p.15, l.12: 'to quantify' -¿ 'for quantifying'*

Fixed.

We are looking forward to your reply,
Mariano Mertens
(on behalf of all co-authors)

**References**

Dunker, A. M., Yarwood, G., Ortmann, J. P., and Wilson, G. M.: Comparison of Source Apportionment and Source Sensitivity of Ozone in a Three-Dimensional Air Quality Model, Environmental Science & Technology, 36, 2953–2964, doi:10.1021/es011418f, URL http://dx.doi.org/10.1021/es011418f, pMID: 12144273, 2002.

Karamchandani, P., Long, Y., Pirovano, G., Balzarini, A., and Yarwood, G.: Source-sector contributions to European ozone and fine PM in 2010 using AQMEII modeling data, Atmospheric Chemistry and Physics, 17, 5643–5664, doi:10.5194/acp-17-5643-2017, URL https://www.atmos-chem-phys.net/17/5643/2017/, 2017.

Kwok, R. H. F., Baker, K. R., Napelenok, S. L., and Tonnesen, G. S.: Photochemical grid model implementation and application of VOC, $NO_x$, and $O_3$ source apportionment, Geoscientific Model Development, 8, 99–114, doi:10.5194/gmd-8-99-2015, URL http://www.geosci-model-dev.net/8/99/2015/, 2015.

Li, Y., Lau, A. K.-H., Fung, J. C.-H., Zheng, J. Y., Zhong, L. J., and Louie, P. K. K.: Ozone source apportionment (OSAT) to differentiate local regional and super-regional source contributions in the Pearl River Delta region, China, Journal of Geophysical Research: Atmospheres, 117, doi:10.1029/2011JD017340, URL http://dx.doi.org/10.1029/2011JD017340, d15305, 2012.

Valverde, V., Pay, M. T., and Baldasano, J. M.: Ozone attributed to Madrid and Barcelona on-road transport emissions: Characterization of plume dynamics over the Iberian Peninsula, Science of The Total Environment, 543, Part A, 670 – 682, doi:http://dx.doi.org/10.1016/j.scitotenv.2015.11.070, URL http://www.sciencedirect.com/science/article/pii/S0048969715310500, 2016.

---

## Author Comment (AC4) · 16 Sep 2019

Dear referee#2 ,
thank you very much for your review of our manuscript GMD-2019-07. Please find our replys to your comments below. In the following, referee comments are given in italics, our replies are in normal font, and text passages which we included in the text, are in bold.

*This manuscript explores whether the source apportionment of surface ozone would be affected by model resolution. It performed model simulations with the different resolution of the model itself and the emission inventories. The difference in the source apportionment using a self-consistent tagging method is attributed to the model resolution and emission inventory resolution. The topic itself and the self-consistent tagging method are interesting.*

Reply: We thank referee#2 for this summary and honouring our work with the self-consitent tagging method.

*However, the analyses presented in the manuscript are too useful;*
Reply: We do not understand this comment.

*the discussion and conclusions are not insightful (or not having any new results as pointed out by Anonymous Reviewer#1). I'd suggest the following items to improve the manuscript.*
Reply: First of all we thank referee#2 for the ideas on how to improve the manuscript which we comment in detail below. As already discussed in our reply the referee#1 we think that we provide new results, because, at least to our knowledge, the impact of the model resolution (and other technical factors) on the results of source apportionment methods has not been investigated in detail. Such an investigation, however, is important for two reasons:

- To investigate how robust the source apportionment results from global models are on the regional scale, and

- to estimate the range of uncertainties of source apportionment caused only by technical limitations of the models and emission inventories.

Even tough our results are only valid for a specific model, they provide new insights about possible ranges on model caused uncertainties. Such results are important for the community involved in source apportionment methods, both on the global and the regional scale.
Finally, we would like to remark that publications in GMD are not primarily about presenting new scientific results. Publications in GMD are mainly to document model developments, document experimental set-ups of model simulations, document evaluation of model systems, present model evaluation strategies and to present technical analyses of model systems. We think that our manuscript documents the influence of model and emission inventory resolutions on source attribution results. This is clearly important to asses source

apportionment results and their related uncertainties, also for other model systems.

To underline the importance of our findings to other modelling communities we largely rewrote the conclusion section. The new part reads:

**Apart from many model specific findings of this study, its results have important implications for other modelling studies and modellers applying source apportionment methods. These implications are:**

- **First, our study shows that average continental contributions of anthropogenic emissions are quite robust with respect to the used model and the used model resolution. This means that global models at coarse resolution can be used to perform ozone source apportionment in this global context.**

- **Second, our results also show that on the regional scale, the differences either caused by different models, but also by model resolution are much larger. These effects arise mainly near hotspot regions like the Po Valley or near major shipping routes in the Mediterranean Sea. However, especially in these areas, contribution analyses of anthropogenic emissions are very important and spurious effects, such as artificially increased ozone levels and contributions caused by the coarse resolution of models and or emission inventories should be avoided. Hence, for regional analyses fine resolved models and emission inventories are required.**

- **Third, our results clearly indicate how large the spread between models with respect to STE is. The importance of stratospheric ozone, both in the global and the regional model, corroborates the necessity of tracing the contributions of stratospheric ozone to ground level ozone explicitly by the source apportionment methods. However, only few currently available methods used on the regional scale account for this process.**

*1. better defining the differences between simulations/models, be specific about what processes causing the variations in source apportionment. Here are just a few examples to improve.*
Reply: As discussed in detail below we think that we are discussing a lot of processes causing these variations in detail. We agree that some explanations could be improved (see below). To better define the model and simulation differences we revised the manuscript accordingly and added the Section 2.2 in which we discuss the different simulations and the motivation for performing these simulation in more detail.

*(a) Meteorological inputs such as temperature and light are different in some simulations, which would result in different biogenic emissions in methods of*

*'on-line calculated' and 'calculated by EMAC.'*

Reply: No: As stated on p5l32ff of the original manuscript MECO(n) is applied in the so called quasi-chemistry transport model mode (QCTM-mode). In this mode the coupling between chemistry and dynamics is disconnected and each model instance simulated the same meteorology in all simulations. Of course, the dynamics differs between the different model instances due to different resolutions and/or physical parametrizations, which leads to differences in the biogenic emissions. We have discussed this issue on p3l4ff (of the original manuscript). For this reason the simulation *EBIO* is performed to investigate the impact of different biogenic emissions.

We added a note about QCTM in Sect. 2.2:
In this mode chemistry and dynamics are decoupled to increase the signal-to-noise ratio for small chemical perturbations. **This means, that even tough the emissions differ in the different simulation each model instance (EMAC, CM50 and CM12) simulated the same dynamics in all simulations. The dynamics between EMAC, CM50 and CM12, however, differs due to different resolution and physical parameterisations.**

Further, we added a longer description on the motivation of the *EBIO* simulation:
**In the simulation *EBIO* the biogenic $C_5H_8$ and soil-$NO_x$ emission as calculated by EMAC are transformed down and applied in CM50. By comparing the results from CM50 of *REF* and *EBIO* the effect of the different biogenic emissions can be analysed. These differences of the biogenic emissions are due to differences in the simulated meteorology between EMAC and CM50.**

*(b) Same anthropogenic emissions in different resolution might result in the same total emission but large regional differences. How do these emissions differ?*

Reply: The coarse resolution of the emissions leads to a dilution of emissions over larger areas. Please see Fig S1 showing the MACCity land transport emissions in EMAC and in CM50. This figure is also added to the revised Supplement. Further, we added tables with the total emissions of the different simulations to the Supplement (Table S2-S10 in the new Supplement). To investigate the impact of the emission resolution onto the results the simulation *ET42* was performed.

*(c) What are actually causing the differences in STE flux in the coarse vs. fine resolution model? Could it be related to on-line vs. off-line meteorology/convections and/or temporal and horizontal averaging of meteorological inputs (just some examples I am familiar with, like in Yu et al. (2018) and Hu et al. (2017); certainly, many other literature on this topic are available)? The*

[Figure]

Figure S1: Annual averaged emissions flux (in molec m$^{-2}$ s$^{-1}$) of NOx due to all anthropogenic emission sources (land transport, anthropogenic non-traffic, shipping; *REF* simulation) for EMAC and CM50.

*contribution from downward transport seems to be the largest differences among models, and it should be quite interesting to explore.*

Reply: Indeed, the largest differences between EMAC and CM50 are the differences of the STE. EMAC and COSMO/MESSy are chemistry-climate models, no chemistry transport models. Hofmann et al. (2016) already investigated in detail differences of the STE between EMAC and COSMO/MESSy, therefore we don't want to discuss this topic in detail again. Generally, the finer resolution of COSMO/MESSy leads to a better representation of the physical processes of individual STE events. However, in our manuscript we do not focus on individual events but rather on multi-year average values. For these multi-year average values the increased contribution of ozone from stratospheric origin is mainly confined to the planetary boundary layer. The reason for this is more efficient vertical mixing in COSMO/MESSy, partly caused by more vigorous convection and by an too unstable boundary layer during night. Taking the biases compared to observations into account this vertical mixing in COSMO/MESSy seems to be too strong, which indicates that the larger contribution of stratospheric ozone (and also for the categories aviation, lightning and N2O) is an artefact of this too strong vertical mixing. As discussed below we added a new Sect. 3 including a model evaluation to the manuscript. Further, we discuss the reason for the STE difference in more detail in the revised manuscript (see various changes in Sect. 4)

*(d) It looks like the total lightning NOx emissions are the same across simulations, do their also have the same 3D distribution?*
Reply: Yes. Over all simulations and over all model instances the same lightning-NO$_x$ emissions are calculated. These are the emissions calculated by EMAC which are transformed during runtime from the EMAC grid onto the grid of CM50/CM12. The procedure is described in the model description section, but we rephrased the description to make it more clear. The new sentence is:
**The lightning NO$_x$ emissions are calculated only in EMAC using a**

**parametrization based on Price and Rind (1992), which is scaled to a global nitrogen oxide emission rate of $\approx 5$ Tg(N) a$^{-1}$ from flashes. In CM50 and CM12 we use the emissions from EMAC (i.e. with same geographical, vertical and temporal distribution), which are transformed on-line onto the grids of CM50 and CM12, respectively.**

*'Inter-model differences' should be better defined and documented and can provide insights on the calculated contributions. Specific discussion of these processes rather than vaguely saying because of the resolution would make this paper more useful.*

Reply: The manuscript is about discussing the processes and other possible explanations for the differences between the different model results. Some examples are (page and line number refer to the original manuscript):

- p9l5ff [..] Due to increased vertical mixing in CM50 compared to EMAC ozone which is produced in the upper troposphere is transported downward more efficiently. [..]

- p10l12ff [..] mainly caused by a decreased net ozone production and a stronger vertical mixing in CM50 compared to EMAC. [..]

- p11l3ff [..] As the analyses of the *ET42* simulation results shows, the coarse resolution leads to an artificial increase of $P_O3$ which in turn leads to an increase [..]

- p11l10ff [..] On the coarse EMAC grid most parts of Southern Italy are considered as sea, affecting especially the calculation of dry deposition in EMAC, as dry deposition of ozone is lower over sea as over land.[..]

- p11l24ff [..] Especially in Southern Germany this is mainly caused by the better resolved topography and larger contributions of stratospheric ozone [..]

- p11l30ff [..] in Western Germany CM12 simulates a larger contribution of the $CH_4$-category to ozone compared to CM50, which is consistent with the larger tropospheric oxidation capacity in CM12 compared to CM50 (Mertens et al., 2016). [..]

We used the term 'inter-model differences' in some parts of the original manuscript to refer to the differences which we discussed before. In some parts we also referred to previous findings of Mertens et al. (2016). We rephrased these parts to be more precise. As an example we added the following note in the newly added Sect. 2.2:
**Differences between the results of the EMAC and CM50 (and CM12) can be attributed to different effects: First, the dynamical core and physical parametrizations between EMAC and COSMO/MESSy differ, second the resolution of these models differs and third EMAC**

**and COSMO/MESSy calculate different soil-$NO_x$ and biogenic $C_5H_8$ emissions. The latter due to the meteorology dependence and due to different soil types in EMAC and COSMO/MESSy.**

Similarly, we added more detailed explanations in Sect. 3 and Sect. 4: An example from Sect. 3:

**As already noted by Mertens et al. (2016), CM50 exhibits a larger positive ozone bias compared to EMAC. This bias is mainly caused by a more efficient vertical mixing in COSMO, as well as by a less stable boundary layer during night. The latter is a common problem of many models leading to diurnal cycles with too large ozone values during night, which results in an overall ozone bias (e.g. Travis and Jacob, 2019). The coarser resolution of the emissions (*ET42*) as well as the different biogenic emissions (*EBIO*) between EMAC and CM50 contribute only partly to the bias of CM50 compared to EMAC. The CM50 ozone bias is larger in *ET42* and *EBIO*. The pattern of the ground level ozone mixing ratio bias of CM50 compared to EMAC is similar for all simulations (see Fig. 3). Generally, CM50 has a positive ozone bias compared to EMAC over most parts of Europe.**

Further, we add some more discussion about the differences of the stratospheric contribution between CM50 and CM12. These differences can mainly be attributed to stronger vertical mixing caused by stronger updraft and downdraft massfluxes in CM12 compared to CM50.

*2. the terms used in the manuscript are very confusing for readers from outside the MESSy model community, particularly when referring to the specific simulation. For example, CM50 is used to compare with EMAC, while one refers to the resolution of 50km of one model; the other refers to a different model. ET42 refers to 'the MACCity emissions are transformed to the coarse grid of EMAC (T42), to investigate the impact of the resolution of the emission inventory.', but it sounds like it is done by the COSMO model only, so do all the REF, EBIO, EVEU simulations. Table 2 seems to suggest that EMAC also has those four simulations. Table 1 is not useful in the context of this manuscript but just adds confusions by adding a bunch of acronyms. This manuscript should not be 'read very much like a technical report for MESSy users' as pointed by the other reviewer. Readability should be improved.*

Reply: We have th feeling that some of the confusion is caused by a missunderstanding of the concept of the MECO(n) model system. However, as the concept is explained in detail in a series of 5 different papers cited in Sect. 2 we wanted to recap only the basic concept of MECO(n). Obviously this basic recap was too short. Therefore we added a slightly longer description of MECO(n). This new part reads:

 **We apply the MECO(n) model system, which couples the global chemistry-climate model EMAC during runtime (i.e. on-line) with**

the regional chemistry-climate model COSMO-CLM/MESSy (Kerkweg and Jöckel, 2012b). Both models, EMAC and COSMO-CLM/MESSy, calculate the physical and chemical processes in the atmosphere and their interactions with oceans, land and human influences. They use the second version of the Modular Earth Submodel System (MESSy2) to link multi-institutional computer codes (Jöckel et al., 2010). The core atmospheric model of EMAC is the 5th generation European Centre Hamburg general circulation model (ECHAM5, Roeckner et al., 2006). The core atmospheric model of COSMO-CLM/MESSy is the COSMO-CLM model (Rockel et al., 2008), a regional atmospheric climate model jointly further developed by the CLM-Community based on the COSMO model. In the model systems acronym 'n' denotes the number of COSMO-CLM/MESSy instances nested into the global model framework. The initial and boundary conditions, which are required for each of these nested regional model instances, are provided by the next coarser resolved model instance. This model instance can either be EMAC or COSMO-CLM/MESSy. Due to the on-line coupling the boundary conditions for the regional model instances can be provided at every time step of the driving model instance. This especially important to resolve short term variations of chemically active species. As EMAC and COSMO-CLM/MESSy calculate both, atmospheric dynamics and composition, the meteorological and chemical boundary conditions are as consistent as possible. In addition, the same chemical solver and kinetic mechanism is applied, leading to highly consistent chemical boundary conditions. Therefore, there is no need of lumping (i.e. treading different chemical species with similar chemical formula as one species), scaling boundary conditions for specific chemical species or taking boundary conditions from different models. More details about the MECO(n) model system are presented in a set of publications including a chemical and meteorological evaluation (Kerkweg and Jöckel, 2012a,b; Hofmann et al., 2012; Mertens et al., 2016; Kerkweg et al., 2018). The set-up of the simulation applied in the present study is very similar to that described by Mertens et al. (2016). Therefore, we present only the most important details of the model set-up. The complete namelist set-up is part of the Supplement.

It is important to understand that in every model simulation different instances of the MECO(n) model run at the **same** time and share necessary boundary and initial fields via MPI communication. For the applied MECO(2) set-up the running model instances are: EMAC, COSMO/MESSy with 50 km resolution (named COSMO(50km)/MESSy) and COSMO/MESSy with 12 km resolution (named (COSMO(12km)/MESSy)). These terms where introduced by Hofmann et al. (2012) and to ease readability the short terms CM50 and CM12 were introduced by Mertens et al. (2016). We don't want to add confusion by introducing new terms and therefore stick to these previously introduced abbreviations. We think that this is the best way of having a clearly defined model system. However, we are open for concrete suggestions for an improved naming.

To make clear thatin MECO(n) different model **instances** run at the same time (and we do not perform simulations with different models) we used the term troughout the revised manuscript.

Using the MECO(2) system (EMAC $\rightarrow$ CM50 $\rightarrow$ CM12) we performed different simulations (*REF*, *EVEU*, *EBIO* and *ET42*) and compare the results of all three model instances. To differentiate between model instances and simulation names, the simulation names are written in italics throughout the manuscript. As shown in Table 2, EMAC is running in all simulations, but with the same set-up including the same emissions. To make this more clear we added a new subsection called 'investigation concept' (Sect. 2.2).

With respect to Table 1 we do not agree with referee#2. We think this table is the best and shortest way of showing the model set-up, which is (to our opinion) very important in terms of reproducibility. We list the name of the individual submodels as they are published in peer reviewed literature under these names. For people not familiar with these submodels we have a short description, stating the physical/chemical process or the diagnostic provided by this submodel as well as a reference describing the individual submodel in detail.

*3. this paper could benefit from a section of model evaluation by adding comparisons with observations. This way could suggest which simulations are 'in practice' better and if the model simulations are actually realistic.*

Reply: We added a basic section of model evaluation and compare the performance of the individual model instances for the different simulations with observations (new Sect. 3). For this, we use ground based station measurements as well as ozone sonde measurements. This should give an impression of the overall model performance. However, from this model evaluation it is not possible to evaluate ozone contributions as these are pure model diagnostics.

*4. the metrics used to quantify simulation difference: this manuscript mostly uses the average concentration of ozone and relative contribution of a specific source. These tend only to show minimal differences among simulations; even though the manuscript claims 'up to 20%' in the calculated contribution of transport emissions, the absolute amount is small. One way to improve is looking at the probability distribution of concentrations or contributions, which could be much more useful to examine differences in model chemical pathways and for specific air pollution episodes, i.e., examples like in Fiore et al. (2002) and Yu et al. (2016).*

Reply: We do not fully agree with this comment. In Fig. 3 and Fig. 7 we show box-whisker plots which indicate the range of the simulated values (e.g.

[Figure]

Figure S2: 95th percentile of the contribution of $O_3^{tra}$ to ground level $O_3$ (for JJA between 9–18 UTC) for **(a)** EMAC, **(b)** CM50 and **(c)** CM50 transformed onto the EMAC grid (CM50$_E$).

range, 25th and 75th percentile, mean and median). Of course we could also have chosen PDFs instead, but they offer similar information. Further, we also discuss differences of the 95th percentile of the contributions of land transport. The differences of these extreme values are of course larger than the average differences.

We agree that the differences for these averaged values show only small differences, but the focus of your analysis is on time scales on which global models (e.g. multi-year averages) and not on the scale of individual pollution events. Therefore, we prefer to stick to the applied metrics. Further, we are not aware where we claim 'up to 20%' in the calculated contributions of transport emissions', as we claim a differences of up to 20 % between the simulated contribution of the different models/model set-ups. These differences are simulated around the Naples region, were the relative contributions between EMAC (17 %) and CM50 (13 %) differ. These relative contributions refer to absolute contributions of 3 to 4 nmol mol$^{-1}$. For the 95th percentile (see Fig. S2) of the relative contribution of $O_3^{tra}$ these difference increase to around 6 percentage − points. We added this figure to the Supplement (Fig S6).

We are looking forward to your reply,

Mariano Mertens
(on behalf of all co-authors)

**References**

Hofmann, C., Kerkweg, A., Wernli, H., and Jöckel, P.: The 1-way on-line coupled atmospheric chemistry model system MECO(n) Part 3: Meteorological evaluation of the on-line coupled system,

Geosci. Model Dev., 5, 129–147, doi:10.5194/gmd-5-129-2012, URL http://www.geosci-model-dev.net/5/129/2012/, 2012.

Hofmann, C., Kerkweg, A., Hoor, P., and Jöckel, P.: Stratosphere-troposphere exchange in the vicinity of a tropopause fold, Atmospheric Chemistry and Physics Discussions, pp. 1–26, doi:10.5194/acp-2015-949, URL https://doi.org/10.5194/acp-2015-949, 2016.

Jöckel, P., Kerkweg, A., Pozzer, A., Sander, R., Tost, H., Riede, H., Baumgaertner, A., Gromov, S., and Kern, B.: Development cycle 2 of the Modular Earth Submodel System (MESSy2), Geosci. Model Dev., 3, 717–752, doi:10.5194/gmd-3-717-2010, URL http://www.geosci-model-dev.net/3/717/2010/, 2010.

Kerkweg, A. and Jöckel, P.: The 1-way on-line coupled atmospheric chemistry model system MECO(n) Part 1: Description of the limited-area atmospheric chemistry model COSMO/MESSy, Geosci. Model Dev., 5, 87–110, doi:10.5194/gmd-5-87-2012, URL http://www.geosci-model-dev.net/5/87/2012/, 2012a.

Kerkweg, A. and Jöckel, P.: The 1-way on-line coupled atmospheric chemistry model system MECO(n) - Part 2: On-line coupling with the Multi-Model-Driver (MMD), Geosci. Model Dev., 5, 111–128, doi:10.5194/gmd-5-111-2012, URL http://www.geosci-model-dev.net/5/111/2012/, 2012b.

Kerkweg, A., Hofmann, C., Jöckel, P., Mertens, M., and Pante, G.: The on-line coupled atmospheric chemistry model system MECO(n) – Part 5: Expanding the Multi-Model-Driver (MMD v2.0) for 2-way data exchange including data interpolation via GRID (v1.0), Geoscientific Model Development, 11, 1059–1076, doi:10.5194/gmd-11-1059-2018, URL https://www.geosci-model-dev.net/11/1059/2018/, 2018.

Mertens, M., Kerkweg, A., Jöckel, P., Tost, H., and Hofmann, C.: The 1-way on-line coupled model system MECO(n) – Part 4: Chemical evaluation (based on MESSy v2.52), Geoscientific Model Development, 9, 3545–3567, doi:10.5194/gmd-9-3545-2016, URL http://www.geosci-model-dev.net/9/3545/2016/, 2016.

Price, C. and Rind, D.: A simple lightning parameterization for calculating global lightning distributions, J. Geophys. Res. Atmos., 97, 9919–9933, doi:10.1029/92JD00719, URL http://dx.doi.org/10.1029/92JD00719, 1992.

Rockel, B., Will, A., and Hense, A.: The Regional Climate Model COSMO-CLM (CCLM), Meteorol. Z., 17, 347–348, 2008.

Roeckner, E., Brokopf, R., Esch, M., Giorgetta, M., Hagemann, S., Kornblueh, L., Manzini, E., Schlese, U., and Schulzweida, U.: Sensitivity of Simulated Climate to Horizontal and Vertical Resolution in the ECHAM5 Atmosphere Model, J. Climate, 19, 3771–3791, doi:10.1175/jcli3824.1, URL http://dx.doi.org/10.1175/jcli3824.1, 2006.

Travis, K. R. and Jacob, D. J.: Systematic bias in evaluating chemical transport models with maximum daily 8 h average (MDA8) surface ozone for air quality applications: a case study with GEOS-Chem v9.02, Geoscientific Model Development, 12, 3641–3648, doi:10.5194/gmd-12-3641-2019, URL `https://www.geosci-model-dev.net/12/3641/2019/`, 2019.

---

## Referee Report (RR1)

The focus of the paper is to study the impact of i) model resolution ; ii) the resolution of the anthropogenic emissions inventory ; and iii) the inventory itself on the contributions of different emission sources through a source apportionment method. This is done by several simulations, where a different factors is modified each time in an effort to disentangle the effects. Modelled results are compared between simulations to understand the sensitivity of the system to each factor. Lots of very interesting points are raised in the analysis of the results. However, before publication, I think that the paper needs to attend to several issues. There are also, many errors in the English that create confusion and even make it impossible to understand certain sentences.

**General Comments**

Several simulations are conducted to try and disentangle the different effects. I am not convinced that the set-up of the model experiments is the most appropriate to thorough investigate the issues raised here. The results should read more like suggestion of potential effects rather than quantified impacts. Moreover, to make their points clear, the authors need to restructure the presentation of their results. The analysis of the different simulations should follow a more systematic pattern. This would help the reader understand which effect is studied at each section by comparing which simulations?

More specifically, all results in section 3 are based on comparisons between the EMAC and CM50 grids. First of all, the difference between these tow resolutions is huge (300km vs. 50km). To properly investigate the effect of the resolution, as done in the papers cited by the authors, intermediate grid resolutions should be used in my opinion. Furthermore, why do results of the CM12 simulation are not discussed in section 3 (Table 3) ? One would expect a more meaningful comparison between the CM50 and CM12 model resolutions, where meteorology is more consistent. Why EVEU simulations between CM50 and CM12 are never discussed?

The effect of the resolution of emissions is studied, but no explanation is provided on how emission inventories are applied over different resolution grids. What proxies are used, what assumptions are made? It would make much more sense to apply a first projection of emissions on the finer grid (CM12) with whatever proxies and then just add up emissions to the coarser grids with no further assumptions. This is not what is done here. Moreover, a different emission inventory is applied for the EVEU simulation but the authors don't discuss nor show at all in what this inventory is more accurate.

The authors suggest that the enhanced overestimation of ozone at CM50 compared to coarser resolutions is due to enhanced vertical mixing during night-time. Wouldn't it be interesting instead of averaging so much in time to look at how well or bad the diurnal cycles are represented?

Some important points need to be discussed more. Ozone formation requires both NOx and COV and the ozone production depends on the ratio between these concentrations in a non-linear way. NOx and COV are emitted by different sources (road transport is mainly responsible for NOx emissions while COV are largely emitted by biogenic activity). It is therefore, not straightforward how the source apportionment works to simulate the contributions of these sources to ozone formation. Of course this is explained in previous publications but still, in my point of view, a brief explanation would improve the understanding of the presented work.

I think that the concept of using lateral and top boundary conditions for the contributions to ozone production needs to be discussed in more detail. It does not seem straight forward to me how boundary conditions of source apportionment could be applied. The contribution of an emission source to ozone production should depend on the chemical regime (i.e. the ration of NOx over COV emissions). These ratios depend strongly on the size of the grid cells as well as what sources happen to be included in the corresponding volumes. They are very resolution dependent. Imagine a large city: one grid configuration may have a grid-cell including the entire city another configuration may cut it in two and dilute the cities emissions with "cleaner air" from rural areas. The ozone production will be radically different in the two configurations. If we add ozone production from the different configurations over the same areas covering the city and the surrounding rural areas the ozone production over the same areas will not match. Consequently

source apportionment will not give the same contributions. How then could contributions estimated over larger grid cells are applied as boundaries for smaller grid-cells. I am sure reasonable assumptions are done here. They should be discussed in the paper in my opinion.

**Specific Comments**

-Some possible reasons for the differences in the results of the simulations are mentioned in the paragraph ln 10 of page 8. It is not clear if ozone concentrations or contributions or both are discussed here?

-I know this issue was raised at the previous phase of the review but I still find the terminology unnecessarily confusing. In a few lines at the introduction we see all these models.

page 2 ln 24: MECO(n) = MESSy-fied ECHAM and COSMO shouldn't it be "or COSMO"
Page 2 ln 26: EMAC = ECHAM5/MESSy
Page 2 ln 27: COSMO/MESSy

What is the difference between MESSy-fied ECHAM and ECHAM5/MESSy?
A few lines later the terms COSMO-CLM/MESSy appears (what is CLM?), MESSy2 and ECHAM5.

If I understand correctly the idea here is that for global-scale modelling the ECHAM climate model is coupled (on-line) with the chemical mechanism of the MESSy chemistry transport model and the result is called EMAC. For regional-scale modelling the regional scale model MESSy is coupled on-line or forced (please specify) with the regional scale meteorological model COSMO.

The coupling of both systems I.e ECHAM and COSMO/MESSy is called MECO.

A figure showing all this might be helpful.

-The number of vertical layers is provided for CM12 and 50 (40 layers up to 22km) and EMAC (31 layers up to 10hp). It would be helpful to provide both heights on the same unit (either pressure or height or both). It would also be important to provide the first layer's thickness since ground level ozone is of interest here.

-S4: Figure caption goes up to k sub-plots but the figure only has up to h sub-plots.

-There are lots of spelling and grammar errors in the manuscript. I suggest some corrections bellow but they are not exhaustive.

Page 1, Ln 19: a large difference
Page 1, Ln 20: role
Page 3 ln 22 This especially... The sentence is incomplete
Page 4 ln 2: CM12 not C12.
Page 8 ln 3: The set-ups of… (were applied) is varied… There is some problem here.
Page 8 ln 13: don't all three EMAC, CM50 and CM12 have the same emissions in ET42?
Page 10 ln 2: root mean square error
Page 10 ln 2: over an area
Page 15 ln 25 the numbers of relative contributions given here for EMAC and CM50 do not match Figure 7b.
Page 17 ln 13: as for the mean values
Page 19 ln 14 ..stays similar

---

## Author Response (AR2)

Dear editor

Thank you very much for guiding through the editorial process. We are very sorry about the typos in the public discussions manuscript. In the revised manuscript we therefore carefully checked and improved its language.

We appreciate the recommendations from referee #3 which helped to improve the manuscript. In detail we changed the following things:

- We added more details describing the source apportionment method.
- We added a short description of the regridding transformation of the emissions.
- We added a comparison of the CM12 results with observations to the supplement.
- To reduce the confusion with the different model acronyms we used the term "COSMO-CLM/MESSy" throughout the manuscript.

Some of the recommendations from referee #3, however, are beyond the scope of the manuscript. Therefore after careful consideration, we decided to abstain from changing the manuscript further with respect to these comments. For further details see the reply to referee #3.

Attached are the comments to the two referees (original comments in italic, answers in normal fonts, changes in the manuscript in bold) together with the revised manuscript. In the revised manuscript all modifications are highlighted (latexdiff).

We are looking forward to your reply,

Mariano Mertens
(on behalf of all co-authors)

Dear referee#1

Thank you very much for your review of our manuscript GMD-2019-07. Please find our replies to your comments below. In the following, referee comments are given in italics, our replies in normal font, and text passages which we included in the text are in bold.

*The authors have adequately addressed my concerns and I now recommend publication of the paper.* Reply: Thanks for the recommendation.

*The paper needs careful proof-reading as there are a number of typos, many of which should be caught with a spell-checker. For example, the last sentence of the abstract should have 'show large' instead of 'show a large', 'role' instead of 'ole', and 'explicitly' instead of 'explicily'. Some typos will not be caught by a spell-checker, so careful reading is required.*
Reply: Indeed, we have overlooked these typos. We are very sorry for this. We carefully checked the manuscript before resubmission.

*Such as:*
*p.2, l.8: 'tough' - 'though'*
*p.3, l.22: 'This' - 'This is'*
*p.3, l.26: 'treading' - 'treating'*

Reply: Thanks. We fixed these errors.

Thanks again,
Mariano Mertens
(on behalf of all co-authors)

Dear referee#3

Thank you very much for your review of our manuscript GMD-2019-07. Please find our replies to your comments below. In the following, referee comments are given in italics, our replies in normal font, and text passages which we included in the text are in bold.

*The focus of the paper is to study the impact of i) model resolution ; ii) the resolution of the anthropogenic emissions inventory ; and iii) the inventory itself on the contributions of different emission sources through a source apportionment method. This is done by several simulations, where a different factors is modified each time in an effort to disentangle the effects. Modelled results are compared between simulations to understand the sensitivity of the system to each factor. Lots of very interesting points are raised in the analysis of the results. However, before publication, I think that the paper needs to attend to several issues. There are also, many errors in the English that create confusion and even make it impossible to understand certain sentences.*

Reply: We thank referee#3 for this summary. Please see below for a more detailed response to the raised issues. For the new submission we checked the manuscript carefully for spelling and grammar errors.

*Several simulations are conducted to try and disentangle the different effects. I am not convinced that the set-up of the model experiments is the most appropriate to thorough investigate the issues raised here. The results should read more like suggestion of potential effects rather than quantified impacts. Moreover, to make their points clear, the authors need to restructure the presentation of their results. The analysis of the different simulations should follow a more systematic pattern. This would help the reader understand which effect is studied at each section by comparing which simulations?*

Reply: In the conclusion we state very clearly that the quantified values of course are only valid for the chosen set-up/model. Here we write:
"For the specific model set-up involving the global model EMAC and the regional model COSMO-CLM/MESSy our results show that simulated differences of ozone [...] "
However, as also stated in the conclusion, we think that the overall implications of the study are important for other studies:
"Apart from many model specific findings of this study, its results have important implications for other modelling studies and modellers applying source apportionment methods. These implications are: [...]"
Besides this we are generally very thankful and open for any constructive criticism. The comment of referee#3, however, criticises our structure of our manuscript vaguely without given concrete recommendations. While drafting the manuscript we chose the current structure of the manuscript for the following reasons: First we wanted to discuss differences of the ozone production between EMAC and CM50, because differences of the ozone production are a

main reason for differences of the contributions. After this we compare the ozone contributions as simulated by EMAC and CM50 averaged over Europe, before we compare the differences exemplarily for the tagging categories land transport and biogenic in more detail. To disentangle the differences between the contributions simulated by EMAC and CM50 we discuss in this context the results of the specific sensitivity studies.

*More specifically, all results in section 3 are based on comparisons between the EMAC and CM50 grids. First of all, the difference between these tow resolutions is huge (300km vs. 50km). To properly investigate the effect of the resolution, as done in the papers cited by the authors, intermediate grid resolutions should be used in my opinion. Furthermore, why do results of the CM12 simulation are not discussed in section 3 (Table 3) ? One would expect a more meaningful comparison between the CM50 and CM12 model resolutions, where meteorology is more consistent. Why EVEU simulations between CM50 and CM12 are never discussed?*

Reply: Indeed, the difference between 300 km and 50 km are quite large. This jump of the resolution is a typical jump going from global chemistry-climate models to regional chemistry-climate models as for instance applied in classical off-line nesting / downscaling approaches. As the focus of our manuscript is the question if the results from global and regional models are comparable we think this jump in the resolution is justified to investigate the scientific question addressed in our manuscript (i.e. What is the difference of ozone source apportionment results between global an regional models?).

We do not discuss the results of CM12 in Sect. 3 as the domain of CM12 does only cover Germany. The model domain of CM12 corresponds only to very few gridboxes in EMAC and a comparison between EMAC and CM12 is not meaningful. In order to not add more information (and possible confusions) to the manuscript we didn't add the comparison of CM50 and CM12 to the manuscript either. However, we now added the scores of CM50 and CM12 for the *REF* and the *EVEU* simulation to the Supplement (Table S2). Further, it is important to note that we compare the model results only to stations from EMEP (which are mainly 'background' stations), as our primary focus is not on the air quality scale.

We do not agree that the differences between CM50 and CM12 for *EVEU* are never discussed. Starting on page 12 l 15 onwards of the revised manuscript we discuss the results of *EVEU* for CM50 and CM12. Figure 9 shows the comparison between CM50 and CM12 for *EVEU*.

*The effect of the resolution of emissions is studied, but no explanation is provided on how emission inventories are applied over different resolution grids. What proxies are used, what assumptions are made? It would make much more sense to apply a first projection of emissions on the finer grid (CM12) with whatever proxies and then just add up emissions to the coarser grids with no further assumptions. This is not what is done here. Moreover, a different emis-*

*sion inventory is applied for the EVEU simulation but the authors don't discuss nor show at all in what this inventory is more accurate.*

Reply: We are not using any proxies for applying the emission inventories at the different resolutions. We use the emission inventories in the resolution at which they are provided by the original source, meaning with a resolution of $0.0625°$ for the VEU emission inventory and 0.5 x 0.5° resolution for the MACCity emission inventory (cf. Table 2). The emission inventories are online regridded (using a conservative regridding approach) onto the grid of the respective model. This regridding is done by the MESSy submodel GRID described by Kerkweg et al. (2018). Thus, indeed we do exactly what referee#3 is asking for, the fine grained emissions are transformed (conservatively) to the coarser grid without further assumptions. We added a note on this point in the revised manuscript: " For the *REF* simulation the same MACCity inventory is applied in EMAC, CM50 and CM12 at its finest available resolution. This means, that the MAC-City emissions are transformed onto a grid of 2.8 x 2.8° resolution in EMAC and to a grid of 0.44 x 0.44° in CM50 (and 0.1 x 0.1° resolution in CM12). **The transformation from the original resolution of the emissions onto the model grid is performed online (i.e. during runtime) via the MESSy submodel GRID (Kerkweg et al., 2018). Here, a conservative remapping approach is used to transform the emissions onto the model grid. We chose this approach, because in this way we need to store the emission data only once at their original resolution and we are in that way always using the finest possible resolution. Further, it is important to note that we do not use any proxies for the downscaling of the emissions on the model grid (e.g. population density). However, due to the different model resolutions the emissions are distributed differently into the gridboxes.** The different geographical distribution of the emissions due to the transformation onto the finer grids is shown in Fig. S16 in the Supplement). This simulation serves as reference. Differences between the results of the EMAC and CM50 (and CM12) can be attributed to different effects: "

We didn't discuss the details of the VEU emission inventory applied in *EVEU* on purpose. The goal of our manuscript is not to evaluate different emission inventories for Europe. As stated in the revised manuscript (p8l25ff) we apply the changed emission inventory only to set the inter-model differences in the context of the uncertainties caused by different emission inventories. A detailed discussion of the difference between the emission inventories for Europe is beyond the scope of the manuscript. For the detailed discussion of the differences between the emission inventories please see our manuscript Mertens et al. (2019).

*The authors suggest that the enhanced overestimation of ozone at CM50 compared to coarser resolutions is due to enhanced vertical mixing during night-time. Wouldn't it be interesting instead of averaging so much in time to look at how well or bad the diurnal cycles are represented?*

Reply: Of course one could also investigate diurnal cycles of ozone, as we have done in Mertens et al. (2016). The focus of our manuscript, however, is more on climatological scales and monthly average values than on values for distinct periods of the day. In follow up studies one could of course investigate diurnal cycles, daily maxima or other ozone metrics in more detail. In this manuscript this is beyond scope.

*Some important points need to be discussed more. Ozone formation requires both NOx and COV and the ozone production depends on the ratio between these concentrations in a non-linear way. NOx and COV are emitted by different sources (road transport is mainly responsible for NOx emissions while COV are largely emitted by biogenic activity). It is therefore, not straightforward how the source apportionment works to simulate the contributions of these sources to ozone formation. Of course this is explained in previous publications but still, in my point of view, a brief explanation would improve the understanding of the presented work.*

Reply: We added information on how the source apportionment method is working, but as the method is described in detail by Grewe et al. (2017) and Grewe (2013) we don't want to repeat all details of the source apportionment method. The changed paragraph reads:

"We apply the TAGGING submodel described by Grewe et al. (2017) for source apportionment. The tagging method is a diagnostic method, i.e. the atmospheric chemistry calculations are not influenced. Due to constraints with respect to the computational resources (e.g., computing time and memory), the detailed chemistry from MECCA is mapped on a family concept, for which the tagging is performed. The tagged species are ozone, the family of $NO_y$, the family of NMHC, CO, PAN as well as OH and $HO_2$ in a steady state approach. **The tagging method itself is based on the combinatorical ansatz described by Grewe (2013). In the tagging concept the mixing ratios of the considered chemical species and families are fully decomposed into $N$ unique categories, meaning that the sum of mixing ratios over all considered categories equal the total mixing ratio of the considered species (i.e. the budget is closed):**

$$\sum_{\text{tag}=1}^{N} O_3^{\text{tag}} = O_3. \tag{1}$$

**As an example for the generalised tagging method we consider the production of ozone by the reaction of NO with an organic peroxy radical ($RO_2$) to $NO_2$ and the organic oxy radical (RO):**

$$NO + RO_2 \longrightarrow NO_2 + RO. \tag{2}$$

**According to Grewe et al. (2017) (Eq. 13 and 14 therein) this leads to the following fractional apportionment:**

$$P_2^{\text{tag}} \quad = \tfrac{1}{2}P_2 \left( \frac{\text{NO}_y^{\text{tag}}}{\text{NO}_y} + \frac{\text{NMHC}^{\text{tag}}}{\text{NMHC}} \right). \tag{3}$$

**$P_2$ is the production rate of $O_3$ by reaction 2. $\text{NO}_y$ and NMHC are the mixing ratios of the corresponding tagged families, while species marked with $^{\text{tag}}$ represent quantities tagged for a specific category (e.g., stratosphere, land transport, etc.). The denominator represents the sum of the mixing ratios over all categories of the respective tagged family/species. Accordingly, the tagging scheme takes into account the specific reaction rates from the full chemistry scheme. Further, the fractional apportionment is inherent to the applied tagging method as due to the combinatorical ansatz, every regarded chemical reaction is decomposed into all possible combinations of reacting tagged species."**

*I think that the concept of using lateral and top boundary conditions for the contributions to ozone production needs to be discussed in more detail. It does not seem straight forward to me how boundary conditions of source apportionment could be applied. The contribution of an emission source to ozone production should depend on the chemical regime (i.e. the ration of NOx over COV emissions). These ratios depend strongly on the size of the grid cells as well as what sources happen to be included in the corresponding volumes. They are very resolution dependent. Imagine a large city: one grid configuration may have a grid-cell including the entire city another configuration may cut it in two and dilute the cities emissions with 'cleaner air' from rural areas. The ozone production will be radically different in the two configurations. If we add ozone production from the different configurations over the same areas covering the city and the surrounding rural areas the ozone production over the same areas will not match. Consequently source apportionment will not give the same contributions. How then could contributions estimated over larger grid cells are applied as boundaries for smaller grid-cells. I am sure reasonable assumptions are done here. They should be discussed in the paper in my opinion.*

Reply: Generally, model results are only valid for the resolution at which the model is operated. Also $O_3$ or $NO_2$ mixing ratios or the representation of convection heavily depend on the model resolution. As discussed by referee#3 also the results of source apportionment methods depend on the resolutions. Exactly this point is the motivation for this study. We wanted to investigate how the source apportionment results depend on the resolution. Our manuscript clearly shows that on the continental scale the differences caused by the resolution are not very large. This shows that source apportionment results of a global model can be used as lateral boundaries conditions for the regional model. Technically, the "contributions" at the lateral boundaries of the regional model instances are prescribed by the corresponding tracers, not as fractions. We slightly rephrased the corresponding paragraph in the manuscript to make this more clear:

"The TAGGING submodel is applied in each model instance. At the lateral and top boundaries of CM50 and CM12 the tagged contributions are treated in the same manner as all chemical species, i.e. the mixing ratios of the tagged species **of the finer model instance (i.e., the absolute contributions)** are relaxed towards the mixing ratios **of the tagged species** provided by the driving model instance. "

*Some possible reasons for the differences in the results of the simulations are mentioned in the paragraph ln 10 of page 8. It is not clear if ozone concentrations or contributions or both are discussed here?*
Reply: This paragraph describes general reasons for differences of the simulation results from EMAC and CM50. To make this more clear we revised the sentence: **Differences of the simulation results (i.e. mixing ratios, contributions and dynamics) of EMAC and CM50 (and CM12) can be attributed to different effects:**

*I know this issue was raised at the previous phase of the review but I still find the terminology unnecessarily confusing. In a few lines at the introduction we see all these models.*
*page 2 ln 24: MECO(n) = MESSy-fied ECHAM and COSMO shouldn't it be 'or COSMO'*
*Page 2 ln 26: EMAC = ECHAM5/MESSy*
*Page 2 ln 27: COSMO/MESSy*
*What is the difference between MESSy-fied ECHAM and ECHAM5/MESSy?*
*A few lines later the terms COSMO-CLM/MESSy appears (what is CLM?), MESSy2 and ECHAM5.*
Reply: The long name of MECO(n) as given in the manuscript is correct, as it describes the model system consisting of ECHAM, MESSy and COSMO. Indeed, we agree that the terms MESSy-fied ECHAM and ECHAM5/MESSy introduce some confusion, as both describe the same model system. Due to historical reasons these two abbreviations exist and are used in the literature. To reduce the different acronyms a little bit we now use the term COSMO-CLM consistently trough-out the whole manuscript. COSMO-CLM is the COSMO model in regional climate model mode, developed by the CLM-Community.

*If I understand correctly the idea here is that for global-scale modelling the ECHAM climate model is coupled (on-line) with the chemical mechanism of the MESSy chemistry transport model and the result is called EMAC. For regional-scale modelling the regional scale model MESSy is coupled on-line or forced (please specify) with the regional scale meteorological model COSMO.*
Reply: No. The modelling framework is the second version of the Modular Earth Submodel System (MESSy2, Jöckel et al., 2010). **MESSy is no chemistry transport model!** MESSy is a middle-ware providing model infrastructure (such as memory management, timer and a tracer infrastructure) and several so called submodels. These submodels are process descriptions (either chemical or physical) or diagnostics (such as the TAGGING). MESSy is integrated in

several base models (ECHAM5, CESM1, COSMO-CLM, ICON). In the context of this manuscript ECHAM5/MESSy (aka EMAC) and COSMO-CLM/MESSy are of importance. Both models are chemistry-climate models and consists of the dynamical cores (ECHAM5 and COSMO) and descriptions of chemical and dynamical processes. The descriptions of the physical processes (and the dynamical cores) of EMAC and COSMO-CLM/MESSy differ, the chemical process descriptions are the same. These chemical processes are described in several submodels (e.g., for chemical kinetics, dry deposition etc.). COSMO-CLM/MESSy is coupled on-line to EMAC allowing for a seamless 'zooming' into the results of the global model via the regional model (this we call 'on-line nesting').

To our understanding we describe this very clearly at the beginning of Sect. 2: "We apply the MECO(n) model system, which couples the global chemistry-climate model EMAC during runtime (i.e. on-line) with the regional chemistry-climate model COSMO-CLM/MESSy (Kerkweg and Jöckel, 2012b). Both models, EMAC and COSMO-CLM/MESSy, calculate the physical and chemical processes in the atmosphere and their interactions with oceans, land and human influences. They use the second version of the Modular Earth Submodel System (MESSy2) to link multi-institutional computer codes (Jöckel et al., 2010). The core atmospheric model of EMAC is the 5th generation European Centre Hamburg general circulation model (ECHAM5, Roeckner et al., 2006). The core atmospheric model of COSMO-CLM/MESSy is the COSMO-CLM model (Rockel et al., 2008), a regional atmospheric climate model jointly further developed by the CLM-Community based on the COSMO model."

*The coupling of both systems I.e ECHAM and COSMO/MESSy is called MECO.* Reply: This is almost correct. It is ECHAM5/MESSy (aka. EMAC) and COSMO-CLM/MESSy.

*A figure showing all this might be helpful.* Reply: Figure S1 gives an overview about all these components. However, the MECO(n) system is documented in detail in a series of five publications (Kerkweg and Jöckel, 2012a,b; Hofmann et al., 2012; Mertens et al., 2016; Kerkweg et al., 2018). This manuscript is not intended to serve as an additional documentation of the model system. Therefore, we don't like adding this figure to the manuscript as it would introduce additional complexity and acronyms which need to be discussed.

*-The number of vertical layers is provided for CM12 and 50 (40 layers up to 22km) and EMAC (31 layers up to 10hp). It would be helpful to provide both heights on the same unit (either pressure or height or both). It would also be important to provide the first layer's thickness since ground level ozone is of interest here.* Reply: ECHAM5 uses the pressure as vertical axis, while COSMO (in the chosen set-up) uses a geometrical height axis. Therefore the vertical axis can not be directly compared. However, we added approximate geometrical heights for ECHAM5 in the manuscript:

[Figure]

Figure S1: Sketch of the MECO(n) systems. The basemodels ECHAM5 and COSMO are coupled to MESSy. MESSy provides the process descriptions (e.g. ONEMIS, JVAL) and diagnostics (TAGGING) as submodels. The nesting of COSMO in ECHAM5 or itself is performed using the MESSy submodel MMD* which takes care of the communication via point to point MPI communication. The transformation of the boundary conditions onto the COSMO grid is performed by INT2COSMO (I2C) which is a part of the submodel MMD*.

"Both COSMO-CLM/MESSy instances use **40 vertical model levels (terrain following) with geometric height as vertical coordinate. The model top is at a height of $\approx$ 22 km, the damping zone starts at 11 km height. The thickness of the lowest model layer is $\approx$ 20** m. The boundary conditions for CM50 are provided by EMAC, which is operated at a resolution of T42L31ECMWF, i.e. with a spherical truncation of T42 (corresponding to a quadratic Gaussian grid of approx. 2.8° x 2.8° in latitude and longitude) with 31 hybrid pressure levels in the vertical up to 10 hPa **(corresponding to around 30 km over Europe). The thickness of the lowest model layer corresponds over Europe to $\approx$ 60** m."

*-S4: Figure caption goes up to k sub-plots but the figure only has up to h sub-plots.*
Reply: Thanks. Fixed!

*-here are lots of spelling and grammar errors in the manuscript. I suggest some corrections bellow but they are not exhaustive.*
Reply: We are sorry for the typos and very thankful for the given corrections. We corrected the mistakes and checked the manuscript very carefully.

*Page 1, Ln 19: a large difference*
Reply: Fixed

*Page 1, Ln 20: role*
Reply: Fixed

*Page 3 ln 22 This especially? The sentence is incomplete*
Reply: Fixed

*Page 4 ln 2: CM12 not C12.*
Reply: Fixed

*Page 8 ln 3: The set-ups of? (were applied) is varied? There is some problem here.*
Reply: Fixed. Is now: **The set-ups of the CM50 instance and CM12 instance (if applied) are varied systematically between the different simulations**

*Page 8 ln 13: don't all three EMAC, CM50 and CM12 have the same emissions in ET42?*
Reply: CM12 was not active for the simulation *ET42*. The resolution of the emissions are much too coarse to gain meaningful results with CM12.

*Page 10 ln 2: root mean square error*
Reply: Fixed

*Page 10 ln 2: over an area*
Reply: We guess page 10 line 19 was meant here. We fixed it.

*Page 15 ln 25 the numbers of relative contributions given here for EMAC and CM50 do not match Figure 7b.*
Reply: Yes, because different metrics are used. Figure 7 shows averages over the CM50 domain for the ozone column up to 850 hPa for 2008–2010. In the text we give the relative contributions to ground level ozone for a specific region (defined as rectangular box from 10° W: 30° E and 32° N: 65° N) for 2008. In the text we give the values for 2008 only, because the sensitivity studies have been performed for 2008 only.

*Page 17 ln 13: as for the mean values*
Reply: Fixed

*Page 19 ln 14 ..stays similar*
Reply: Fixed

We are looking forward to your reply,
Mariano Mertens
(on behalf of all co-authors)

**References**

[revised manuscript text omitted]